# G4mer: An RNA language model for transcriptome-wide identification of G-quadruplexes and disease variants from population-scale genetic data

Farica Zhuang[1], Danielle Gutman[2], Nathaniel Islas[1], Bryan B. Guzman[3], Alli Jimenez[4], San Jewell[2], Nicholas J. Hand [2,5], Katherine Nathanson [6,7], Daniel Dominguez [3,4,8] & Yoseph Barash [1,2] ✉

RNA G-quadruplexes (rG4s) are key regulatory elements in gene expression, yet the effects of genetic variants on rG4 formation remain underexplored. Here, we introduce G4mer, an RNA language model that predicts rG4 formation, classifies rG4 subtypes, and evaluates the effects of genetic variants across the transcriptome. G4mer significantly improves accuracy over existing methods and uncovers subtype-specific differences in mutational sensitivity and evolutionary constraint, highlighting sequence length and flanking motifs as important rG4 features. Applying G4mer to 5′ untranslated region (UTR) variations, we identify variants in breast cancer-associated genes that alter rG4 formation and validate their impact on structure and gene expression. These results demonstrate the potential of integrating computational models with experimental approaches to study rG4 function, especially in diseases where non-coding variants are often overlooked. To support broader applications, G4mer is available as both a web tool and a downloadable model.

RNA G-quadruplexes (rG4s) are secondary structures formed in guanine-rich regions of RNA, which have emerged as crucial regulatory elements in gene expression. A canonical RNA G-quadruplex consists of four runs of at least three guanines (G-runs) separated by 1-7 nt loops of any nucleotide $(G_{3+}N_{1-7})_3G_{3+}$. The four G-runs associate via Hoogsteen hydrogen bonds to form planar G-tetrads, which then stack atop one another and are stabilized by monovalent cations (e.g., $K^+$) (Fig. 1a). These structure motifs are particularly enriched in UTRs of mRNAs[1,2], where they play key roles in various regulatory processes including translation, alternative splicing, polyadenylation, and mRNA

stability[3–7]. Alterations to rG4 structures have been linked to human pathologies, such as cancer and neurological diseases, by disrupting the gene expression landscape in affected individuals[8]. For example, an intronic RNA G-quad has been shown to affect CD44 exon 8 inclusion through interactions with RNA-binding protein (RBP) hnRNPF, leading to inhibition of epithelial to mesenchymal transition (EMT) and the associated cell migration and invasion[7]. Another example is the 5′ UTR of the NRAS oncogene where an rG4 was shown to repress translation and represents a potential therapeutic target in cancer[9]. More generally, rG4 alterations have been shown to affect RNA

[1]Department of Computer and Information Science, University of Pennsylvania, Philadelphia, PA, USA. [2]Department of Genetics, Perelman School of Medicine, University of Pennsylvania, Philadelphia, PA, USA. [3]Department of Pharmacology, University of North Carolina at Chapel Hill, Chapel Hill, NC, USA. [4]Department of Biochemistry and Biophysics, University of North Carolina at Chapel Hill, Chapel Hill, NC, USA. [5]Institute for Translational Medicine and Therapeutics, Perelman School of Medicine, University of Pennsylvania, Philadelphia, PA, USA. [6]Division of Human Genetics and Translational Medicine, Dept of Medicine, Perelman School of Medicine, University of Pennsylvania, Philadelphia, PA, USA. [7]Basser Center for BRCA, Abramson Cancer Center, Perelman School of Medicine, University of Pennsylvania, Philadelphia, PA, USA. [8]RNA Discovery Center, The University of North Carolina at Chapel Hill, Chapel Hill, USA. ✉e-mail: yosephb@upenn.edu

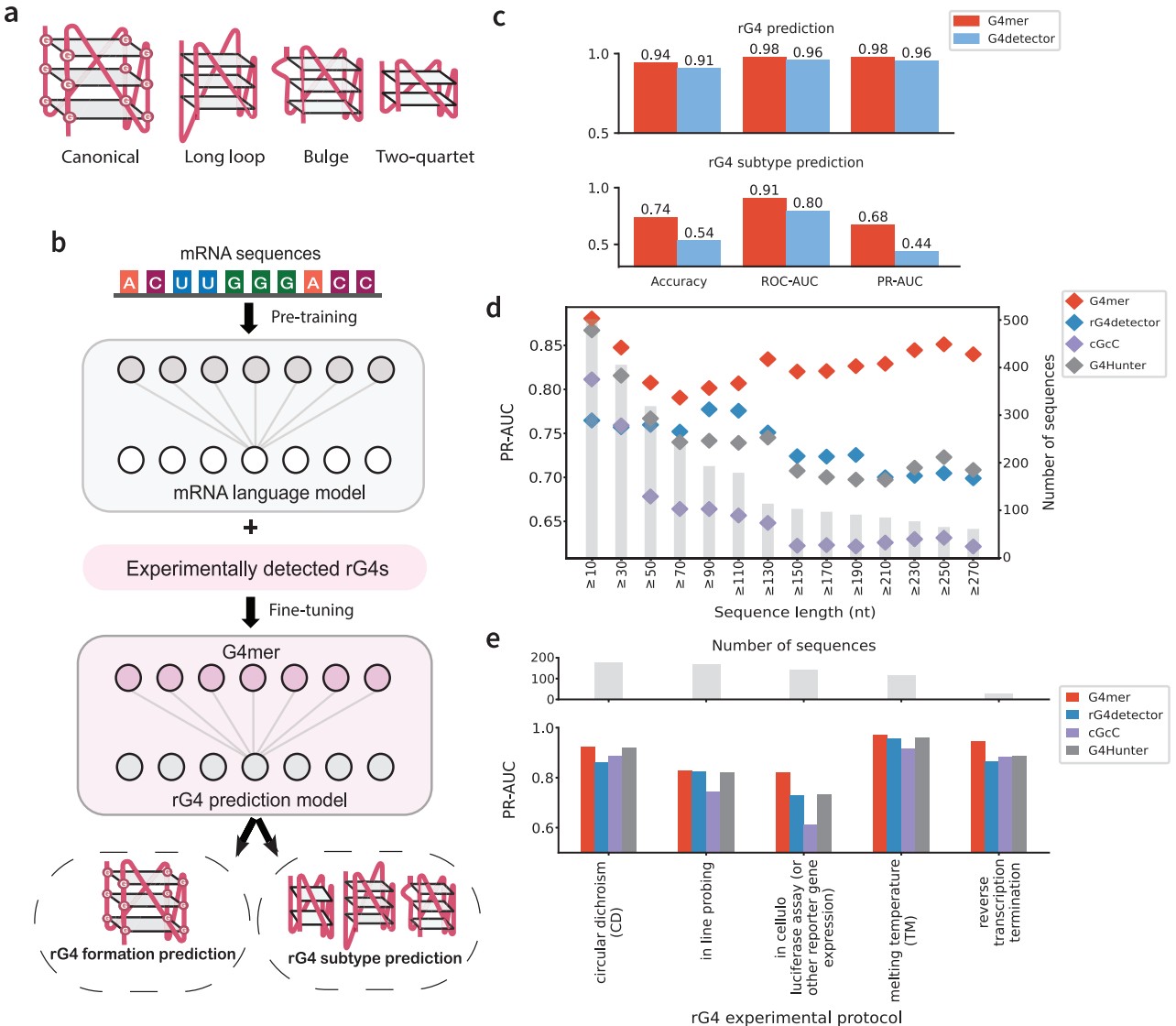

**Fig. 1 | G4mer model overview and performance across multiple benchmarks.**
**a** rG4 structures can be categorized into canonical as well as noncanonical subtypes such as long loop, bulges, and two-quartet. **b** G4mer is developed based on an RNA language model that was pre-trained on the entire human transcriptome and fine-tuned with experimentally detected rG4 sequences. **c** Comparison of transformer-based G4mer (red) and CNN-based rG4detector (blue) for rG4 binary prediction (top) and rG4 subtype multiclass prediction (bottom), evaluated by accuracy, ROC-AUC, and PR-AUC. **d** PR-AUC performance of G4mer (red) compared to rG4detector (blue), cGcC (purple), and G4Hunter (gray) across sequence lengths from the G4RNA database. Gray bars indicate the number of sequences per length bin; colored points show PR-AUC. **e** PR-AUC comparison across the top 5 experimental rG4 detection protocols. The top bar plot shows the number of sequences; the bottom plot shows PR-AUC values across models.

processing such as translation[10–12]. Consequently, rG4s have garnered increasing attention for their therapeutic potential[13–16]. Despite the growing recognition of their importance, the influence of genetic variants on rG4 formation, particularly at the transcriptome-wide level, remains poorly understood.

Recent advancements in high-throughput sequencing, such as rG4-seq[1], have enabled the mapping of rG4s across the transcriptome. However, these methods are often limited by noise, condition specificity, and the inability to capture rG4s in lowly expressed genes[17]. As a result, computational models to predict G4 formation have been developed. Traditional methods are score-based, typically relying on G/C content, skewness, and the number of consecutive Gs present in the sequence to score the propensity of G4s[18–22]. Nevertheless, such methods may overlook noncanonical forms struggle in regions where flanking sequences introduce noise by altering the G/C ratio, leading to inaccurate predictions of rG4 formation. More recently, machine

learning-based models have been developed to predict rG4 formation[23–25]. These models, particularly those based on convolutional neural networks (CNNs)[26–29], have achieved state-of-the-art performance in predicting rG4 structures by learning patterns from labeled datasets. Yet, CNN-based models are inherently limited by the size of their convolutional kernels, which are designed to capture short-range interactions. This limitation may make it challenging for CNNs to accurately predict rG4 formation in longer, transcriptome-wide sequences. Arguably, a more accurate model can not only improve predictions of rG4 formation but also help detect the effects of genetic variants affecting rG4s and gain insights into the sequence features governing rG4 formation. Specifically, distinguishing between rG4 subtypes such as canonical, bulged, two-quartet, and long-loop motifs is important as these exhibit distinct thermodynamic properties and may influence protein binding, folding kinetics, and susceptibility to disruption by genetic variants[11,30].

To create such a model, we developed G4mer, a transformer-based[31] RNA language model designed to predict rG4 formations across the human transcriptome. We evaluated its performance against current state-of-the-art methods using diverse sequences spanning a wide range of sequence lengths, transcriptomic contexts, and experimental methodologies. These datasets include sequences derived from various high-throughput and structure-probing techniques, ensuring a comprehensive assessment of G4mer's predictive capabilities. G4mer is also trained to distinguish between rG4 subtypes, allowing analyses of how different structural topologies relate to stability, disruption, and function.

In this work, we present G4mer, a transformer-based RNA language model for transcriptome-wide prediction of rG4 formation and subtype classification. G4mer outperforms existing methods across diverse sequence lengths, transcriptomic contexts, and experimental datasets, offering strong generalizability. Using G4mer, we predict rG4s across the transcriptome and assess the impact of genetic variants, with a focus on the 5' and 3' UTRs of mRNAs where rG4s are most densely populated[1,2]. Variants predicted to disrupt rG4s are under negative selection, and rG4 length is an overlooked but important factor in rG4 prediction and experimental analysis. We map rG4 subtypes across the transcriptome, identify sequence features contributing to rG4 formation, and link rG4-altering variants from Penn Medicine Biobank (PMBB)[32] to breast cancer through phenome-wide association studies (PheWAS). We validate the rG4-altering effects of selected disease-associated variants by circular dichroism spectroscopy and dual-luciferase assays, confirming their structural and protein expression effects. Finally, we provide an open-access web tool for transcriptome-wide rG4 and genetic variant predictions to facilitate further research.

## Results

### G4mer is a transformer-based model that improves rG4 formation and subtype predictions

To develop G4mer, we first pre-trained mRNAbert, a bidirectional encoder representation model (BERT)[33], using masked language modeling on the entire human transcriptome (see Methods). The pre-training step allowed the model to learn generalized representations of RNA sequences in a self-supervised manner. Following pre-training, we fine-tuned mRNAbert specifically for rG4 prediction using a high-throughput rG4 detection experimental dataset[34] along with non-rG4 sequences (Fig. 1b). For this, we utilized the rG4-seeker dataset, which contains over 5000 regions identified as containing rG4 elements through the rG4-seq[1] in HeLa cells, then post-processed to ensure accurate detection of rG4 sequences[34]. To complement this, an equal number of G-rich negative sequences, sampled from transcripts with no rG4 detected, were included to define the non-rG4 sequences (see Methods for additional details). We first performed standard 10-fold cross-validation (CV) on the dataset of rG4 and non-rG4 sequences to evaluate rG4 prediction performance of G4mer compared to the CNN-based model G4detector[26], which previously achieved state-of-the-art performance in predicting G4 formation. On this task, G4mer outperformed G4Detector by multiple measures of performance: 0.94 vs 0.91 in accuracy, 0.98 vs 0.96 in receiver operating characteristic-area under the curve (ROC-AUC) and precision recall-area under the curve (PR-AUC, see Fig. 1c).

Following this initial comparison, we investigated the models' abilities in predicting rG4 subtypes on the aforementioned rG4-seeker dataset of experimentally identified rG4 sequences. In this data, the authors applied an rG4 sequence classification scheme[1,34] and categorized each sequence into one of eight labels based on sequence motifs and predicted stability, ranging from canonical to known non-canonical subtypes such as long loops, bulges, and two-quartets (Fig. 1a). Other categories account for different G-rich sequence patterns that do not fit predefined classes. The classification process

follows a hierarchical assignment approach. rG4 sequences that matched the pattern of multiple categories were assigned to the class with the higher predicted stability, following the order of canonical rG4s, long loops, bulges, two-quartets, and so on[1] (see Methods). Therefore, identifying the rG4 subtype category of a given sequence relates not only to the sequence patterns and logical rules but also to the associated stability of the sequences. In this context, G4mer significantly outperformed G4Detector, achieving an accuracy of 0.74, ROC-AUC of 0.91, and PR-AUC of 0.68 compared to an accuracy of 0.54, ROC-AUC of 0.8, and PR-AUC of 0.44 for G4Detector (Fig. 1c). G4mer's improved subtype classification demonstrates its ability to capture intricate sequence patterns and implicitly assess the stability of the sequences. Overall, these results highlight the robustness of G4mer's transformer-based architecture and its superior predictive capability in identifying rG4 formation, both in general and for specific subtypes.

We next sought to assess the prediction capabilities beyond the specific data and experimental techniques of rG4-seq. We therefore turned to an independent set of 795 published sequences from the G4RNA database[35]. The G4RNA database includes sequences tested under 24 distinct experimental protocols from various publications, spanning RNA structure-probing methods such as SHAPE, SHALiPE, and DMS probing, as well as biophysical techniques like circular dichroism (CD) and melting temperature (Tm) analysis, among others (Supplementary Fig. 2a). The sequences were filtered to ensure consistency, including the removal of sequences reported by multiple sources with conflicting labels (see Methods). We then compared G4mer's performance against CNN-based rG4detector[27], as well as the scoring-based methods cGcC[18] and G4Hunter[19]. G4mer outperformed all the models across all the filtered G4RNA sequences (Supplementary Fig. 1).

Notably, the G4RNA database contains sequences derived from diverse transcriptomic contexts, including artificial constructs, transcript fragments, and full-length UTR sequences, validated using structure-probing, biochemical, and biophysical methods. These sequences vary greatly in length from from 14 to 1368 nt. Although flanking regions are important for accurate rG4 formation prediction, including excessive lengths in the sequence input may introduce noise, for example, by adding additional G-rich elements that can confuse the model and reduce prediction accuracy. When evaluating all sequences (10 nt or longer), G4Hunter performance was close to that of G4mer (Fig. 1d). However, as we focused on progressively longer sequences, a significant performance difference emerged. G4mer remained less sensitive to increasing sequence length, consistently outperforming other models with PR-AUC values around 0.85 for all lengths. In contrast, other models suffered from a noticeable decline in performance (Fig. 1d). This suggests G4mer's robustness in handling longer sequences, making it suitable for transcript-wide prediction analyses, particularly those involving UTR regions that span hundreds to thousands of nucleotides.

To assess G4mer's generalizability to sequences from diverse experimental protocols, we compared the performance of G4mer, G4detector, cGcC, and G4Hunter across the top 5 rG4 experimental protocols in G4RNA (Fig. 1e). These protocols included circular dichroism (CD), in-line probing, in cellulo luciferase assay (or other reporter gene expression), melting temperature (Tm), and reverse transcription termination. Overall, G4mer offers improved performance over existing models across all experimental protocols. While we focus on the five most frequently represented protocols in the dataset, G4mer's improved performance extends across the broader range of experimental protocols in G4RNA, including additional structure-probing methods and more (Supplementary Fig. 1c). Notably, G4mer performs well on sequences from both the CD protocol and the in cellulo luciferase assay, achieving a PR-AUC of 0.82 despite the stark differences in their sequence length distributions (Supplementary Fig. 2b). Specifically, the CD protocol has a median sequence

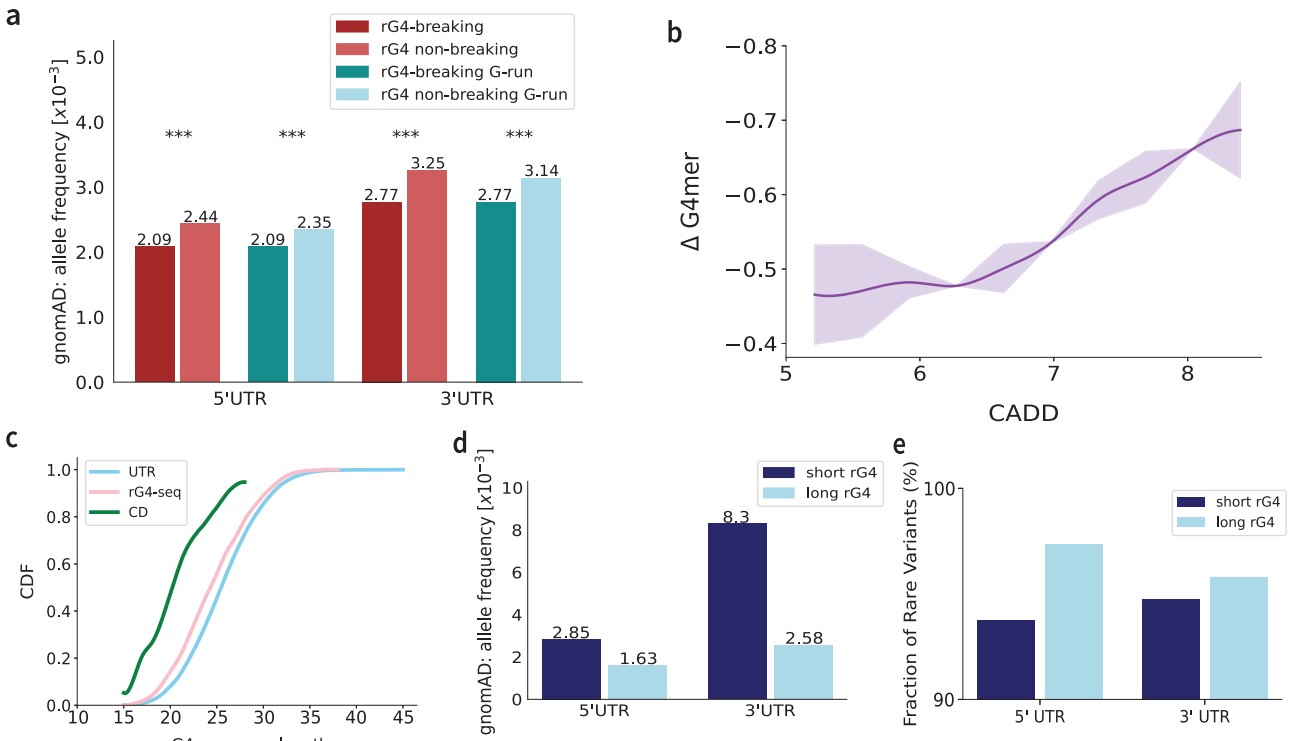

**Fig. 2 | rG4-altering variant analyses. a** gnomAD mean alternate allele frequency comparison between rG4-breaking variants and negative control rG4 variants that are non-breaking across the 5′ and 3′ UTR in all contexts (red) as well as in the G-runs (blue). Asterisks denote significance (p values for the four comparisons are $2.05 \times 10^{-32}$, $1.76 \times 10^{-44}$, $4.25 \times 10^{-16}$, and $7.31 \times 10^{-43}$, respectively, one-tailed KS test). **b** Relationship between raw CADD scores and ΔG4mer values in 5′ UTR regions. ΔG4mer represents the change in predicted rG4 formation probability when comparing sequences with and without a variant. The purple line represents the mean ΔG4mer value across CADD score bins, and the shaded area indicates the standard deviation. **c** Cumulative distribution function (CDF) plot comparing the length distributions of putative canonical rG4s identified in UTR regions (blue), sequences from the rG4-seq experiment conducted on HeLa cells (pink), and sequences that were validated using circular dichrosim (CD) from the G4RNA database (green). **d** Mean alternate allele frequency comparisons of variants breaking short versus long canonical rG4. **e** Fraction of rare variants (MAF < 0.1%) breaking short versus long canonical rG4.

length of 30.5 nt, while the in cellulo luciferase assay has a median of 210 nt (Supplementary Fig. 2c). This demonstrates G4mer's robust ability to accurately predict rG4 formation across diverse experimental conditions, even when faced with varying sequence length distributions that arise from technical differences. These results underscore that, despite being trained on rG4-seq protocol data, G4mer successfully captures the fundamental properties of rG4s, allowing it to generalize well across independent datasets validated through different methodologies and minimize the impact of technical biases.

## G4mer allows transcriptome-wide mapping of rG4-altering variants to assess their functional significance

Having established the high accuracy of G4mer in predicting rG4 formation given a sequence, we leveraged its capabilities to predict the effects of single-nucleotide variants (SNVs) on rG4 formation probability transcriptome-wide. To assess the reliability of these predictions in a population-scale setting, we estimated G4mer's false discovery rate (FDR) using highly expressed genes in HeLa cells. This analysis yielded an FDR of 1.86%, suggesting high specificity for rG4 detection (see Methods). We then applied G4mer to all single-nucleotide variants observed in the human population, as cataloged in the gnomAD database[36], focusing on those located in predicted rG4-forming regions (wild-type G4mer score > 0.7). Specifically, variants that significantly lower the predicted rG4 probability (to < 0.3) are classified as rG4-breaking, while those with minimal predicted effect (change in score within ± 0.02) are considered non-breaking. We then analyzed the rG4-breaking and non-breaking sets to detect signals of negative selection.

Negative selection, where deleterious variants are continuously removed from the population, is crucial for maintaining biological function. Signals of negative selection can be identified by the depletion of variants or, conversely, by an unexpected enrichment of rare variants within a specific class. Therefore, to detect negative selection in rG4 elements, we first evaluated the mean alternate allele frequencies (MAFs) of variants predicted to break rG4s (rG4-breaking) and those in rG4s that do not disrupt the structure (rG4 non-breaking) from regions of matched genomic constraints. Genomic constraint was quantified using the constraint Z-score, which reflects the deviation of observed from expected variations in a given genomic region[36]. To account for differences in sequencing depth in the gnomAD whole genome sequencing dataset, we excluded variants with a call rate below 80%. Overall, we found 8785 5′ UTR and 17,181 3′ UTR variants in gnomAD that are rG4-breaking. Conversely, we found 18,049 5′ UTR variants and 30,005 3′ UTR variants in rG4s that have negligible effects on rG4 structure prediction in similarly constrained regions. Notably, rG4-breaking variants show a significant reduction in mean allele frequency compared to variants in rG4s that don't affect the structure ($p << 4.24e-16$, one-tailed KS test) (Fig. 2a). Interestingly, the subset of these variants that break the consecutive G-run section of the rG4 sequences with similar nucleotide contexts also shows a significant reduction in mean allele frequencies for rG4-breaking variants, consistent with the effects of negative selection. Finally, additional analysis revealed a positive correlation between the magnitude of rG4 disruption (more negative G4mer delta scores) and higher Combined Annotation Dependent Depletion (CADD) scores, particularly for variants in the 5′ UTR region (Fig. 2b). This relationship suggests that

variants that significantly disrupt rG4 structures are more likely to be deleterious, supporting the hypothesis that rG4s may play a critical functional role. A similar, albeit more modest, trend was also observed in the 3′ UTR region (Supplementary Fig. 5), further reinforcing the potential regulatory relevance of rG4-breaking variants across UTR contexts.

Building on the variant analyses above, which underscore the functional significance of rG4s, we next aimed to explore whether the importance of rG4s varies depending on their sequence characteristics, such as length. Specifically, our previous results (Fig. 1d, Supplementary Fig. 2) revealed a strong bias in some existing models and experimental protocols for short rG4 sequences. This performance variation raises the question of whether such a bias may, in turn, lead to overlooking longer rG4s that have functional significance. Naturally, rG4 subtypes could greatly affect the sequence length. For example, the noncanonical long loops are intrinsically longer than canonical rG4s. To minimize variations in lengths contributed by rG4 subtypes, we first focused on canonical rG4s. We observed that the length distributions of sequences could vary based on the experimental technique used. For instance, rG4-seq tends to capture longer rG4 sequences with a median of 24 nt, compared to a commonly used structure validation method, circular dichroism spectroscopy, with a median of 20 nt (Fig. 2c). Notably, putative rG4s in the UTR regions of the human transcriptome often consist of longer sequences with a median of 25 nt that more closely resemble those detected by the rG4-seq protocol. Moreover, tested sequences with CD have a maximum length of 28 nt, while the longest putative rG4s found in the UTR span 45 nt.

Next, we investigated the MAF of variants located within short (≤22 nt) and long (≥28 nt) rG4 sequences across both 5′ UTR and 3′ UTR regions. Distinct differences emerged between the variants located in short and long rG4 sequences across these regions (Fig. 2d). First, the mean allele frequency for constraint-matched variants breaking long canonical rG4 sequences is lower compared to those within short rG4 sequences. To complement this and confirm the trend holds at the lower tail of the distribution, we also examined the fraction of rare variants, finding that long canonical rG4 sequences harbored a proportion of variants with MAF < 0.1% than short rG4s (Fig. 2e). Although these differences were not statistically significant due to the limited number of variants tested, the observed trend suggests that long rG4s may be equally, if not more, functionally important than short rG4s in the transcriptome.

### G4mer reveals genetic variation differences between G4 subtypes and regulatory signals in UTR regions flanking rG4s

To better understand the diversity and functional consequences of rG4s, we first examined subtype-specific properties across transcriptome-wide UTR predictions. Using our subtype classification model, we annotated each predicted rG4 with one of eight subtype categories, including canonical, long loop, bulges, and two-quartet forms (Supplementary Fig. 3a, b). In our main analyses, we focused on the seven characterized subtypes with defined structural features. These subtypes reflect structural heterogeneity in rG4 formation and differ in their relative prevalence across 5′ and 3′ UTR regions compared to their distribution in the G4mer training data (Fig. 3a). We found a consistent trend across both 5′ and 3′ UTR regions where canonical, long loop, and bulges were much more frequent in the original rG4-Seq training data compared to what we observed across the human transcriptome and for rG4s where breaking mutations occurred (Fig. 3a). In contrast, all subtypes involving rG4 with G-rich regions were significantly over-represented in the transcriptome and especially in regions with rG4-breaking mutations. These shifts in distributions were all statistically significant ($p < 10^{-30}$, Chi-square test for each possible pair of distributions, see Methods), and potentially reflect structure stability differences.

The above analysis prompted us to investigate how rG4-breaking genetic variants quantitatively affect rG4 formation probabilities. Specifically, we analyzed the distribution of ΔG4mer scores (the difference between mutant and wild-type scores) for rG4-breaking gnomAD variants located in the GG dinucleotide context. This analysis, shown in Fig. 3b, indicated significant differences between rG4 subtypes (Welch's ANOVA F=520.9, p < < $10^{-3}$). Furthermore, in both 5′ and 3′ UTR, the effects of rG4-breaking variants on bulges, G-rich rG4s, and two-quartet all differed significantly from the canonical subtype (adjusted p < < $4.4 \times 10^{-4}$, post-hoc pairwise Games-Howell test), with the two-quartet showing the largest mean decrease. These differences suggest heightened mutational sensitivity for those subtypes compared to the canonical rG4s. Notably, the ranking of mutational sensitivity among canonical, long loop, bulges, and two-quartet subtypes recapitulates their expected thermodynamic stability hierarchy used to define these categories in prior work[1,34]. Finally, we note that the ΔG4mer distributions were consistently more negative in the 3′ UTR compared to the 5′ UTR (Fig. 3b left vs right-hand side bars). Such a shift could be due to various factors, such as increased mutational sensitivity of rG4s in the 3′ UTR, differences in regulatory mechanisms across UTRs, compositional differences such as the generally lower GC content of 3′ UTR, or some unknown model bias.

To assess whether ΔG4mer differences are reflected in evolutionary constraint, we compared the MAF of rG4-breaking variants in GG dinucleotide context across subtypes (Fig. 3c). To ensure robust estimates, for each UTR, we only retained subtypes represented by at least 30 rG4-breaking variants. In the 5′ UTR, the two quartet subtypes defined by shorter G-runs of two consecutive guanines instead of three or more had significantly higher MAF than the potential G-quadruplex subtype (Dunn's test with Benjamini-Hochberg correction, $p < 0.05$), consistent with weaker purifying selection. In the 3′ UTR, the potential G-triplex subtype, which has only three G-runs instead of the canonical four, showed higher MAF than potential G-quadruplex, two-quartet, and bulges subtypes ($p < 0.05$ for all pairwise comparisons).

The above results not only indicate that mutational constraints vary between structural subtypes, but also raise the question of whether G-run composition or rG4 length itself contributes to mutational sensitivity. To explore length as a factor, we revisited the analysis of rG4 length and constraint (Fig. 2d, e), expanding the comparison beyond canonical motifs to also include long loop rG4s. Specifically, we redefined long rG4s as those with a length ≥ 31 nt, increasing the separation from short rG4s. Under this refined definition, we again observed that variants breaking long rG4s had significantly lower mean allele frequencies in the 5′ UTR ($p < 0.028$, one-tailed Mann-Whitney U test), with a similar but nonsignificant trend observed in the 3′ UTR. Both regions also showed enrichment for rare alleles in long rG4s(Fig. 3d). Taken together, these results support subtype-specific stability and mutational constraint captured by G4mer and suggest long loops may be more functionally constrained, pointing to their potential regulatory importance.

Building on these observations, we next asked whether flanking sequence context could further modulate rG4 function and stability. To explore the mechanistic basis for flank-specific stability, we analyzed the contribution of flanking sequence features to G4mer's predictions in UTRs across the transcriptome using Enhanced Integrated Gradient (EIG) analysis[37], a method for deep learning model interpretation. EIG quantifies the contribution of each feature, specifically k-mers in this context, to a model's prediction by integrating the gradient along a path from a reference point to a sample of interest. Using EIG, we obtained attribution scores for k-mers of different lengths in the flanking regions of rG4 sequences, identifying those that significantly contributed to the prediction compared to non-rG4 sequences ($p < 0.05$, two-sided t-test, Benjamini-Hochberg FDR-adjusted, see Methods for details). The EIG analysis indicates that the

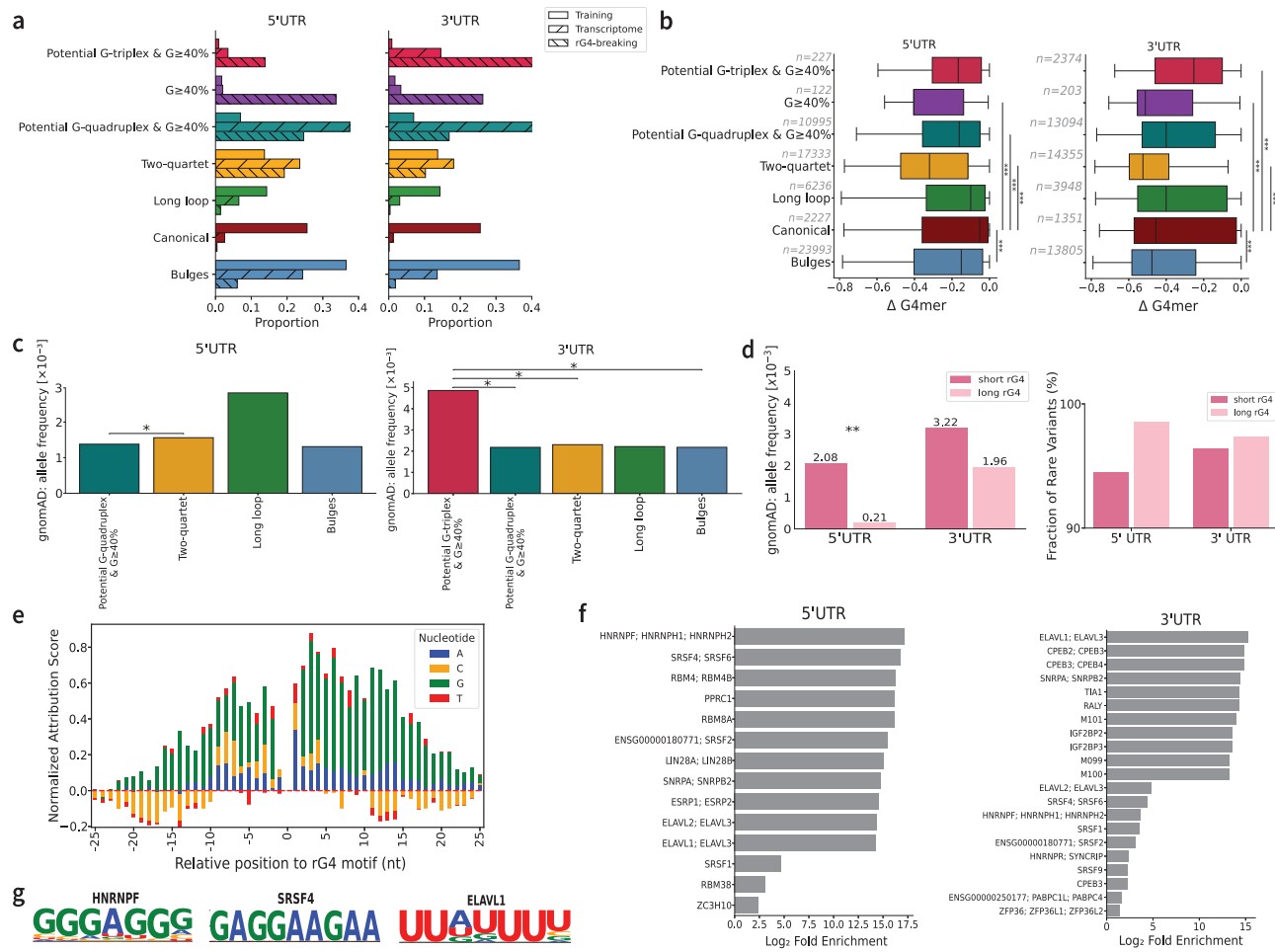

**Fig. 3 | G4mer reveals subtype-specific patterns of evolutionary constraint and interpretable regulatory features in rG4-forming UTRs. a** Distribution of subtypes in the training data, transcriptome-wide, and rG4-breaking regions predicted by G4mer across 5′ UTR and 3′ UTR regions. **b** Distribution of ΔG4mer scores for variants that disrupt rG4s in each subtype. Asterisks denote significance (exact *p*-values from top to bottom: 5′ UTR -- $4.27 \times 10^{-13}$, $4.70 \times 10^{-12}$, $1.58 \times 10^{-12}$, $<1 \times 10^{-300}$; 3′ UTR -- $<1 \times 10^{-300}$, $<1 \times 10^{-300}$, $4.39 \times 10^{-4}$, $<1 \times 10^{-300}$, Welch's ANOVA followed by Games-Howell post hoc). In the box plots, the range of each box extends from the first to the third quartile, with the horizontal line representing the median; whiskers extend to 1.5 times the interquartile range beyond the lower and upper bounds of the box. **c** Bar plots show the mean alternate allele frequency (x10⁻³) for each rG4 subtype in 5′ UTR (left) and 3′ UTR (right). Asterisks denote significant pairwise differences ($p = 1.14 \times 10^{-2}$ for 5′ UTR and $p = 4.66 \times 10^{-2}$ for all significant 3′ UTR comparisons, Dunn's test with Benjamini-Hochberg correction). **d** Comparison of mean alternate allele frequency (left) and fraction of rare variants

(right) for variants breaking canonical and long-loop rG4s. Asterisks denote significance ($p = 0.028$, one-tailed Mann-Whitney U test). **e** Stacked bar plot showing normalized attribution scores for individual nucleotides (A, C, G, T) at each position relative to the rG4 motif across G4mer predicted rG4 sequences transcriptome-wide. Nucleotide-level attribution scores were computed with EIG on the flanking regions of putative canonical rG4 motifs. The x-axis represents positions upstream and downstream of the rG4 motif, with 0 corresponding to the rG4 motif. Positive scores indicate a stronger contribution to rG4 formation, while negative scores suggest reduced influence. **f** RBP motif enrichment in 5′ UTR (left) and 3′ UTR (right) flanking regions of predicted canonical rG4-forming sequences. Enrichment was calculated using motifs from the CISBP-RNA database scanned with FIMO. Only motifs passing a multiple hypothesis-corrected FDR threshold (q < 0.05) and with log₂ fold enrichment > 1 were included. **g** Representative sequence logos for RBP motifs enriched in rG4 flanks, including HNRNPF (left), SRSF4 (middle), and ELAVL1 (right).

sequences flanking the G-runs do indeed contribute to the rG4 predictions (Fig. 3e). Specifically, we find that the positions proximal to the G-runs contribute more to the prediction and that guanine and uracil are preferred in those positions, while cytosine bases generally contribute negatively to predicting rG4 formation. However, the EIG analysis clearly indicates that analyzing attribution per position independently, as is often done when constructing position-specific scoring motifs (PSSM), can be misleading. For example, some cytosines near rG4 motifs showed positive attribution scores, but we observed distinct attribution score patterns for 2-mers and 3-mers (Supplementary Fig. 4a, b). CC dinucleotides consistently had negative attribution scores, whereas dinucleotides containing a cytosine and another nucleotide could show positive attributions. Similarly, trinucleotides with predominantly cytosine bases were negatively

attributed unless other nucleotides were present. In contrast, UU dinucleotides had a strong positive influence on rG4 predictions. Overall, the above results suggest that rG4 formation, as captured by G4mer, avoids competing G-C bonds and prefers open regions via stretches of uridine, emphasizing the role of flanking region composition in rG4 formation. To complement the EIG analysis, which quantified how flanking k-mers contribute to rG4 predictions, we performed a perturbation analysis to directly assess the model's sensitivity to specific nucleotide substitutions in these regions. In this analysis, we systematically mutated guanines and cytosines in the flanking regions and measured the resulting change in G4mer prediction scores (see Methods). Consistent with EIG's positive attribution for guanines and negative attribution for cytosines, we found that G → C substitutions caused the largest decrease in rG4 scores, while

C → G increased them (Supplementary Fig. 4c), reinforcing the role of flanking guanines in stabilizing rG4 structures.

Finally, we turned to analyze whether the guanine- and uracil-enriched flanking regions identified in the aforementioned analysis may serve as binding sites for regulatory proteins. To explore this, we performed motif enrichment analysis using the CISBP-RNA database[38] and FIMO[39], focusing on the UTR flanks of canonical rG4s predicted to fold (G4mer score > 0.7). These were compared to control sequences predicted not to form rG4s (G4mer score < 0.3) (see Methods). Distinct sets of RBP motifs were enriched in rG4 flanks across UTRs (Fig. 3f). In the 5′ UTR, G-binding proteins such as hnRNP F/H and SRSF4/6 were strongly enriched, while in the 3′ UTR, motifs for ELAVL1/3-known to bind U-rich regions, were preferentially enriched (Fig. 3g). Interestingly, uracils are known to stabilize rG4 structures when present in loop regions[30]. Their enrichment in flanking regions suggests potential regulatory coupling via U-binding RBPs, which may similarly contribute to rG4 stability. Moreover, several RBPs such as hnRNP F/H and SRSF4/6 were enriched in both UTR contexts, suggesting shared regulatory mechanisms. Overall, these findings provide mechanistic hypotheses regarding the flanking sequence composition beyond the G-run itself and how these may influence rG4 formation and function via protein-mediated stabilization or modulation.

## Non-coding rG4-altering variants are associated with breast cancer and modulate gene expression

The reduced MAF observed in rG4-altering variants suggests these regions may have functional significance and could play a role in human diseases. To complement these population-level indicators of selective pressure, we next examined whether rG4-breaking effects are associated with clinically annotated pathogenicity. Specifically, we investigated whether ClinVar[40] variants annotated as pathogenic tend to exhibit stronger rG4-breaking effects than those annotated as benign, focusing on variants located in UTRs. We selected variants within high-confidence rG4-forming regions (G4mer score > 0.7), and retained for each variant the transcript with the most disruptive effect (i.e., minimum ΔG4mer score; see Methods, Supplementary Data 1). Overall, we found that variants annotated as pathogenic showed significantly greater rG4-breaking effects, i.e., more negative ΔG4mer scores, than benign variants (Fig. 4a; one-sided Mann-Whitney U test, $p \ll 5.2 \times 10^{-9}$ for 5′ UTR; $p \ll 1.0 \times 10^{-14}$ for 3′ UTR). These results were consistent across both UTR regions and suggest that rG4 disruption may contribute to molecular mechanisms underlying pathogenicity.

Next, to further explore the potential contribution of rG4-altering variants to disease etiology, we focused on breast cancer, a condition where much of the disease heritability remains unexplained[41–43]. While previous work identified an rG4 structure modulating CD44 alternative splicing and EMT transition via hnRNPF binding[7], the extent to which rG4s contribute to post-transcriptional gene regulation in breast cancer remains underexplored. In this study, we focused on rG4-altering variants in UTRs to explore their relevance to breast cancer. Specifically, we performed two types of analysis. First, we conducted phenome-wide association (PheWAS) to test rG4-altering genetic variants for associations with breast cancer. This analysis used naturally occurring SNVs derived from patient exome sequencing data in PMBB (see Methods), enabling us to evaluate potential links between rG4-altering variants and disease risk. Second, we evaluated the patient-derived PMBB variants in 16 genes already implicated in breast cancer for their potential to create or disrupt rG4s in the respective UTR regions. Overall, we identified 28 rG4-breaking variants and 29 rG4-forming variants across eight genes. The full list of variants is included in Supplementary Table 1, with key findings highlighted below.

We detected a significant PheWAS association (p = 7.4e-7) between an rG4-breaking variant in the 5′ UTR of *EPN3* and breast cancer (Fig. 4b). Although *EPN3* was not included in our initial set of breast cancer-associated genes, previous studies have identified it as

an oncogene involved in breast cancer by regulating epithelial-mesenchymal transition, playing a crucial role in metastasis[44]. Furthermore, *EPN3* has been explored as a potential therapeutic target due to its role in regulating apoptosis in breast cancer cells[45]. Hence, the association we observed suggests that rG4-breaking variants may have a broader impact on disease phenotypes and indicate that rG4 structures within the 5′ UTR of *EPN3* may play a critical role in gene regulation. The disruption of rG4s could potentially alter downstream effects, such as gene expression, protein function, or interaction with other cellular components, thereby contributing to cancer susceptibility. Another notable rG4-altering variant was detected in the DNA mismatch repair gene *MSH6*, one of the breast cancer-implicated genes we examined. In this case, the variant was predicted to form, rather than disrupt, an rG4 in the 5′ UTR. *MSH6* has been extensively studied for its associations with Lynch syndrome and its role in increasing the risk of several cancers[46–49].

Given the established role of rG4s within the 5′ UTR in translational regulation[3,10,50], we sought to investigate the functional impact of the aforementioned rG4-altering variants using dual-luciferase reporter assays across three distinct cell lines: NIH3T3, HuH-7, and CHO. For each gene, we constructed two 5′ UTR constructs: the wild-type (WT) sequence and a mutant (Mut) sequence designed to alter the rG4 formation (Fig. 4c, Supplementary Table 2, See Methods). Our results demonstrated a significant modulation of protein expression by rG4-altering variants in both *EPN3* and *MSH6* across all tested cell lines. Specifically, the presence of disease-associated rG4-breaking variants led to a marked reduction in protein expression compared to the wild-type sequences in the case of *EPN3* in all three cell lines (Fig. 4d). Similarly, the rG4-forming variant in *MSH6* consistently resulted in a significant downregulation of protein expression across all cell lines, highlighting the regulatory roles that rG4 structures may play in gene expression (Fig. 4e). While reduced expression of *EPN3* may seem counterintuitive given its reported oncogenic overexpression in breast cancer, this could reflect transcript-specific or cell-type-dependent regulation, warranting further investigation. Overall, the rG4-altering single-nucleotide variants predicted by G4mer induced significant changes to protein expression levels.

## Structural validation of predicted rG4-altering variants with circular dichroism spectroscopy

The observed changes in protein expression due to the predicted rG4-altering variants indicate their functional effects. To explore whether these functional changes were linked to the predicted alterations in rG4 formation, we conducted structural validation using circular dichroism (CD) spectroscopy on RNA sequences from *EPN3* and *MSH6*, both with and without the rG4-altering variants predicted by G4mer. Specifically, we tested the predicted rG4-breaking variant in the *EPN3* and the predicted rG4-forming variant in the *MSH6* RNA sequence (Supplementary Table 3).

Circular dichroism spectroscopy is used to detect the presence and stability of rG4 structures by measuring the differential absorption of left- and right-handed circularly polarized light[51]. G4 structures that fold intramolecularly in a parallel topology, where the G-runs run in the same direction, typically exhibit characteristic CD spectra with a slightly negative peak measured in millidegrees (mdeg) at 240 nm and a positive peak around 260 nm[52] when stabilized by potassium ions (KCl), as potassium specifically supports G4 formation[1,53–55]. Conversely, lithium ions (LiCl) are known to destabilize G4 structures, making them less likely to form or maintain stability under these conditions[1,56,57]. Temperature also plays a critical role; as it is increased from 25°C to 95°C, the thermal energy can create denaturing conditions that disrupt rG4 structures, resulting in a decrease in the characteristic ellipticity at 260 nm[1,58–61]. First, we measured the CD spectra of the wild-type (WT) and mutant *EPN3* RNA under various conditions: 150 mM KCl or 150 mM

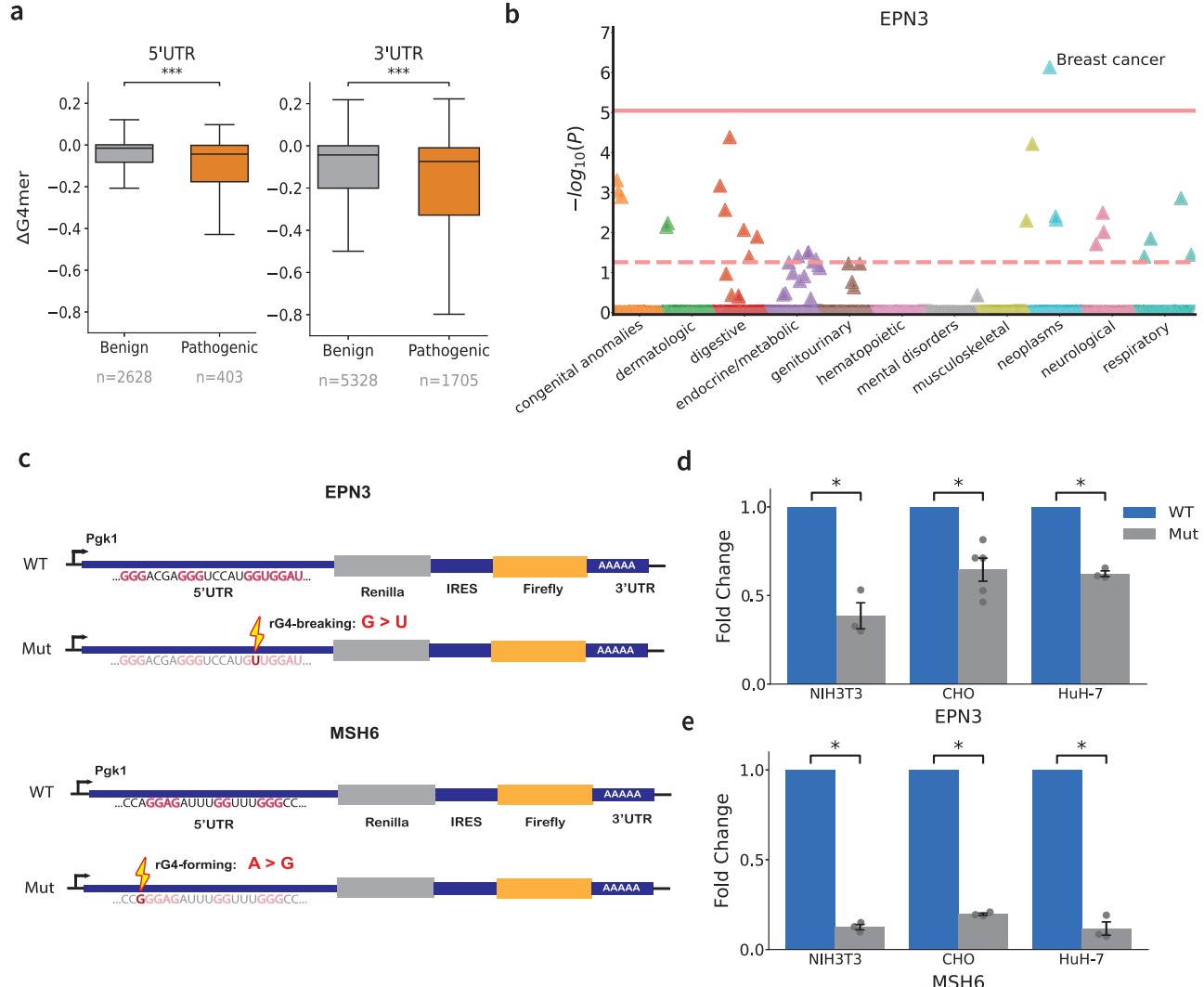

**Fig. 4 | Disease-associated rG4-altering variants change protein expression level. a** Distributions of ΔG4mer scores for ClinVar variants annotated as benign or pathogenic within predicted rG4s in the 5′ and 3′ UTRs. Asterisks denote statistical significance (one-sided Mann-Whitney U test; $p = 5.21 \times 10^{-9}$ for 5′ UTR, $p = 1.03 \times 10^{-14}$ for 3′ UTR). Boxplots show the median (line), interquartile range (box), and 1.5 × IQR (whiskers). Sample sizes: 5′ UTR $n = 2,628$ (benign), $n = 403$ (pathogenic); 3′ UTR $n = 5,328$ (benign), $n = 1,705$ (pathogenic). **b** PheWAS plot for an rG4-disrupting variant in the 5′ UTR of *EPN3*. Each point is a Phecode (ICD-10), colored by disease category and plotted by $-\log_{10}(P)$ from logistic regression adjusted for age, age², sex, and top 10 ancestry PCs. The solid red line indicates the Bonferroni threshold ($p = 8.5 \times 10^{-6}$), and the dashed red line indicates the FDR < 0.1 threshold ($p = 5.5 \times 10^{-2}$). Arrow direction reflects odds ratio: increased risk (up) or decreased risk (down). $N = 1201$ breast cancer cases. **c** Schematic of wild-type (WT) and mutant (Mut) dual-luciferase constructs used to assess the impact of rG4s in the 5′ UTR. WT includes the native sequence; Mut carries single-nucleotide substitutions: **d** rG4-breaking (G > U) in *EPN3* and **e** rG4-forming (A > G) in *MSH6*. Renilla and Firefly luciferase activities were measured to quantify the effect on rG4-mediated translational regulation. Bar heights show mean fold change in Firefly/Renilla activity for Mut vs. WT (normalized to WT = 1), $n = 3$ per condition except in CHO ($n = 5$). For each independent biological replicate, luciferase activity was averaged across ≥3 technical repeats. Statistical significance was assessed using a one-sided Welch's t-test, p-values: *EPN3* -- NIH3T3: $7.00 \times 10^{-3}$, CHO: $2.76 \times 10^{-3}$, HuH-7: $9.16 \times 10^{-4}$; *MSH6* -- NIH3T3: $1.47 \times 10^{-4}$, CHO: $2.79 \times 10^{-5}$, HuH-7: $8.92 \times 10^{-4}$. Error bars represent the standard error of the mean (SEM).

LiCl at both 25 °C and 95 °C. In the rG4-stabilizing KCl buffer at 25 °C, the WT *EPN3* RNA with a predicted rG4 displayed a characteristic positive peak at 260 nm, indicating a stable rG4 structure (Fig. 5a). In contrast, the mutated RNA exhibited a significantly reduced ellipticity at 260 nm ($p < 0.01$), suggesting that the predicted rG4-breaking variant effectively disrupted the rG4 structure. Under the rG4-destabilizing LiCl buffer at 25 °C, the ellipticity of the WT sequence decreased, reaching levels similar to the predicted disrupted rG4 in the mutant sequence. Notably, the mutated RNA maintained a similar ellipticity in both KCl and LiCl buffers, reinforcing the conclusion that the mutant is indeed a disrupted rG4 structure that is unaffected by the different rG4 stabilizing effects of the buffers. At 95 °C, the spectra of both WT and mutant RNAs exhibited a further decrease in ellipticity, consistent with the thermal destabilization of the

rG4 structure. Thus, these results validated the rG4-breaking effect of the variant in *EPN3*.

Next, we measured the CD spectra for the *MSH6* RNA sequence under the same conditions. The spectra of the predicted rG4-forming variant showed an ellipticity at 260 nm that is characteristic of an rG4 under the stabilizing KCl conditions at 25 °C (Fig. 5b). Notably, the WT *MSH6* RNA, which was predicted not to form an rG4, showed a significantly lower ellipticity at 260 nm ($p < 0.01$, independent t-test), suggesting the absence of a stable rG4 structure. In the destabilizing LiCl buffer at the same temperature of 25 °C, the mutant sequence with the predicted rG4-forming variant showed a decrease in ellipticity, suggesting that the predicted rG4 formation was destabilized. There was a further reduction in signals at 95 °C under both buffer conditions, where rG4 structures are expected to be less stable. In all conditions,

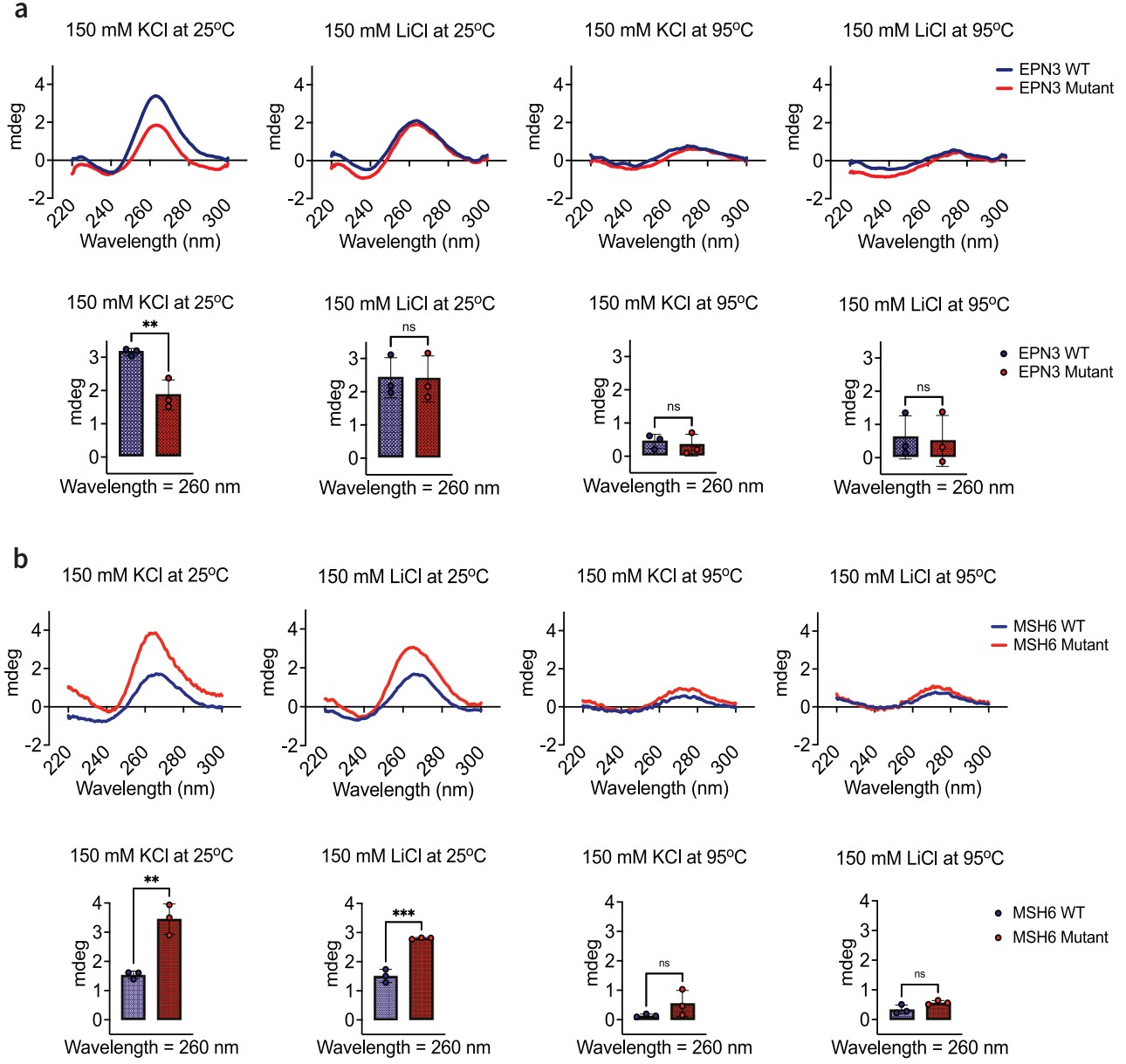

**Fig. 5 | Circular dichroism (CD) spectra validates the predicted rG4-altering effects of variants in *EPN3* and *MSH6*. a** CD spectra measured in ellipticity in millidegrees (mdeg) of the predicted rG4-breaking variant in the *EPN3* RNA sequence (red), compared to the wild-type (WT) (blue), under different buffer conditions (150 mM KCl or 150 mM LiCl) and temperatures (25 °C and 95 °C). Statistical significance is denoted by asterisks (** $p = 8.5 \times 10^{-3}$, independent t-test).

**b** CD spectra of the predicted rG4-forming variant in the *MSH6* RNA sequence (red), compared to the WT (blue), under the same conditions. Each bar plot shows the mean CD signals for the WT and mutant sequences at 260 nm, with error bars representing the standard deviation of three independent replicates per condition. Statistical significance is denoted by asterisks (** $p = 3.6 \times 10^{-3}$, *** $p = 6.0 \times 10^{-4}$, independent t-test).

the WT sequence with no predicted rG4 consistently showed little to no changes in ellipticity, confirming the absence of rG4 in the sequence. Hence, this finding supports G4mer's prediction that the *MSH6* variant promotes rG4 formation in the mutant sequence.

Overall, our structure probing results using CD validate the predictions of G4mer, demonstrating that the *EPN3* variant disrupts an existing rG4 structure, whereby the *MSH6* variant induces rG4 formation, suggesting a link between those structural changes and the observed changes in the downstream gene protein levels measured by the dual luciferase assay.

## Discussion

We introduced G4mer, an RNA language model for predicting rG4s, built on the mRNAbert model, which was trained on the entire human transcriptome. The pre-trained mRNAbert model allows for flexible scaling of the training dataset during fine-tuning to develop G4mer. By leveraging high-throughput rG4-seq data, G4mer outperformed existing state-of-the-art models and demonstrated greater resilience to variations in input sequence lengths. Its robustness to sequence length makes G4mer ideal for both in situ and in vitro studies of RNA G-quadruplex formations, as well as for handling long sequences in transcriptome-wide rG4 predictions.

While existing G4 prediction models have been used to study the effects of variants on rG4 formation[27,62], our study investigates the transcriptome-wide effects of genetic variants on rG4 formation. Variants predicted by G4mer to significantly disrupt rG4s were found to be under negative selection, consistent with previous studies that focused exclusively on canonical rG4s[63]. A key finding from our

analyses of prediction models, experimental datasets, and human UTRs is the effect of rG4 length. Short rG4 sequences have been recognized for their stability[11] and have been the focus of numerous experimental studies. However, our results suggest that longer rG4 sequences, which are underrepresented in both experimental data and algorithmic predictions of rG4s, may be at least equally functionally important and subject to heightened negative selection. One possible explanation for this heightened negative selection in longer rG4s is that their structures are weaker and more flexible, allowing a single nucleotide variation to strongly affect their structure or binding by RBPs that target these regions. This increased vulnerability to genetic variants has been observed in other aspects of RNA processing, such as splicing[64], and warrants further investigation in the context of rG4s. Beyond the G4 motif itself, our expanded analysis further demonstrates that G4mer captures subtype-specific differences. We identified significant differences between the rG4 subtypes distribution in the training data, the transcriptome predictions of subtypes, and the occurrence of variants that disrupt each subtype. We further observed differences between G4 subtypes, which are known to have differences in their stability, in terms of the magnitude of change in G4mer predicted probability upon variant introduction. This result suggests that G4mer may implicitly learn subtype-specific stability differences from sequence data alone, further supporting the biological relevance of its predictions. Furthermore, flanking sequence composition-particularly guanine and uracil content-plays a critical role in G4mer's predictions, as shown by both attribution (EIG[37]) and perturbation analyses. The enrichment of RBP motifs in these regions further points to a model in which flanking sequences help coordinate rG4 formation and regulatory protein binding, which has implications for post-transcriptional control. Our findings align with previous studies on DNA G4s, which highlighted the importance of flanking regions, particularly additional G-runs, in G4 stability[23]. Furthermore, a recent study involving thousands of mutations around several sequences known to form G-Quadruplexes supports our computational model's predictions regarding the contribution of loop length and flanking sequence importance[30].

Our analysis of ClinVar variants highlights the strong potential of further studying rG4 disruption in disease contexts. For example, variants predicted to abolish rG4 structures ($\Delta$G4mer $\leq -0.5$) showed a 5.20-fold enrichment for pathogenicity in the 5′ UTR and a 1.49-fold enrichment in the 3′ UTR (p-values = 2.30e-10 and 8.92e-09, respectively; binomial test). In comparison, SpliceAI-predicted splice-disrupting variants using the recommended 0.2 threshold are even more enriched (12.3-fold in the 5′ UTR and 15.6-fold in the 3′ UTR). However, ClinVar variants are highly biased toward well-established variant classes–particularly those affecting splicing. Thus, we conclude that observing such strong enrichment for a class of variants that has not been systematically studied supports rG4 disruption as an underappreciated mechanism of pathogenicity.

G4mer's application extends beyond merely predicting rG4 formations; it also provides insights into the functional and structural consequences of genetic variants. We found that rG4-altering variants could be associated with diseases. Specifically, in the disease-associated *EPN3* and *MSH6* genes, the respective rG4-breaking and rG4-forming variants led to marked reductions in protein expression compared to the wild-type sequences across multiple cell lines, suggesting that rG4 disruption by genetic variants may contribute to disease phenotypes through altered protein synthesis. Importantly, we showed that the effects of rG4 in the 5′ UTR are context-specific and can either increase or decrease protein expression[5]. On one hand, rG4s can promote translation initiation[65], leading to increased protein levels. On the other hand, rG4s can hinder ribosome scanning[66], thus decreasing protein expression. Our findings reinforce the biological significance of rG4s in translation regulation and their functional importance[67]. For genetic variants, the effects observed in *EPN3* and

*MSH6* support G4mer's ability to inform candidate selection and highlight the functionality of genetic variants in the 5′ UTR.

Similarly, structural validation using circular dichroism (CD) spectroscopy, a primary tool for the characterization of G4 structures[52], confirmed G4mer's predicted structural changes in rG4 formation. The CD spectra demonstrated that the rG4-breaking variant in *EPN3* disrupted the rG4 structure, while the rG4-forming variant in *MSH6* induced the formation of a rG4 structure. However, while the CD spectroscopy results suggest the formation of rG4 structures, we cannot definitively rule out the possibility that these rG4s are forming intermolecularly rather than intramolecularly. This distinction is crucial, as intermolecular rG4s could influence the observed effects differently, potentially impacting the interpretation of their biological significance[68,69]. Nevertheless, these results provide compelling evidence that G4mer can accurately predict the functional and structural impacts of rG4-related variants.

We wish to highlight the following key limitations of the current work. First, unlike our dual luciferase assay experiments, in which the rG4s were surrounded by endogenous flanking sequences, the structure probing experiments lacked this molecular context. Furthermore, we are unable to directly disentangle the potential interplay between sequence, structure, and RBP binding to these elements. Another potential issue relates to the rG4-Seq data used to train G4mer. This data may not capture all rG4s due to sensitivity issues, transient formation, or variability between cell populations, and may suffer from subtype biases. Our study provides functional evidence of rG4-altering variants in two disease-associated genes, demonstrating their impact on post-transcriptional regulation. Broader functional validation across additional disease-associated variants would further refine our understanding of rG4-mediated regulation. Given the low-throughput nature of luciferase reporter assays and circular dichroism spectroscopy, large-scale validation remains a challenge. However, our approach ensures biological relevance by assessing full-length UTR sequences in cellulo, enabling precise quantification of rG4-mediated regulatory effects. Future studies integrating high-throughput functional assays with in-depth mechanistic analyses will help extend these findings across diverse genetic backgrounds and phenotypic contexts. As for the training of the G4mer model, we found that different definitions of the training data can significantly impact model performance. While the positive rG4 sequences in this study were derived from rG4-seq experimental protocols, defining the negative non-rG4 sequences may be done using multiple approaches. In this paper, we enforced a pattern resembling an rG4 in all the negative sequences, but it was not experimentally detected as an rG4 (see Methods). Other models have employed different strategies for defining negative sequences, such as shuffling sequences to preserve the dinucleotide distribution of the positive rG4 sequences or including random windows of non-rG4s[70]. Moreover, to define the positively labeled samples in this study, we treated rG4 occurrences as a classification rather than a regression task, therefore potentially limiting the precision in assessing the effects of genetic variants. Thus, future studies could integrate more diverse datasets and consider the design of training data, which could substantially affect the model's performance and improve generalizability.

We expect that our tool and framework will be valuable for a wide variety of human genetics applications. For instance, G4mer can be applied to analyze ClinVar variants of unknown significance (VUS) or to identify deleterious variants in rare Mendelian diseases. Integrating G4mer scores with additional genomic and clinical data may help to resolve the ambiguity surrounding VUS, a critical need in clinical settings due to the high prevalence of such variants[71,72]. While this study focused on SNVs, future work incorporating phased or multi-nucleotide variants (MNVs) could offer deeper insights into cooperative nucleotide effects on rG4 structures and improve predictive resolution using G4mer. Furthermore, when combined with structural

and functional validations, G4mer has the potential to reveal previously uncharacterized disease associations. Additionally, while G4mer was trained exclusively on human transcriptomic data, future extensions could investigate its applicability to other organisms. rG4s are known to play an important regulatory role in plants, bacteria, and other species[56,73]. With appropriate adaptation, G4mer could be used to investigate species-specific differences in sequence composition, structure stability, and transcriptomic context that influence rG4 formation and regulatory potential. We thus anticipate that G4mer will enable users who are studying rG4 structures to accurately select sequences and rG4-altering variants for further downstream analyses. To support downstream exploration, we provide an interactive web tool (see Code availability section) that enables users to score RNA sequences with G4mer, explore predicted rG4 subtypes, and browse or download transcript- and variant-level predictions.

As genetic studies expand to incorporate more populations, it is crucial that predictive models like G4mer remain applicable across different ancestries. While this study focused on European and African genetic ancestries due to their larger representation in PMBB, the methodological framework we developed is broadly applicable and can be extended to other populations. Our findings demonstrate that rG4 structures influence protein production, reinforcing their role in gene regulation. Expanding this approach to additional ancestry groups will refine our understanding of rG4 formation and its functional consequences across diverse genomic backgrounds. Future studies could leverage large-scale biobanks such as Biobank Japan, KoGES, SG10K Health, GenomeAsia 100K, and other multi-ancestry genomic resources to investigate ancestry-specific rG4-mediated regulatory effects worldwide. As biobank datasets continue to expand, applying our framework across populations will facilitate the identification of previously uncharacterized disease-associated variants and enhance transcriptome-wide predictions of rG4 activity, ultimately advancing our understanding of rG4 function in human health and disease.

More generally, the development and application of G4mer underscores the transformative potential of leveraging language models in RNA biology. Given the results presented in this work, combining an RNA language model with functional and structural experimental methods enables us to gain deeper insights into the functional consequences of genetic variants. In line with the performance of language models in RNA research[74–77], RNA language models stand as a promising approach to study post-transcriptional mechanisms to identify functionally significant variants that may contribute to disease. As we continue to refine these models and integrate them with experimental validations, we anticipate that such tools will become invaluable in studying variant effects in RNA structures and their implications for human health.

## Methods

The Penn Medicine BioBank (PMBB) study, from which human genetic and phenotypic data were obtained, was conducted in accordance with the Declaration of Helsinki and all relevant ethical regulations. PMBB protocols were approved by the University of Pennsylvania Institutional Review Board under the Penn Medicine BioBank (PMBB) protocols (protocol codes: 808346 approved 07/01/2008, 813913 approved 04/03/2013, and 817977 approved 06/06/2013). Access and use of PMBB data for the present study were approved under the University of Pennsylvania IRB protocol 834889. All PMBB participants provided written informed consent for future research use. Analyses were performed on de-identified data to ensure compliance with privacy standards.

### RNA language model

We pre-trained the mRNAbert model based on the BERT architecture[33]. mRNAbert is trained on the sequences in the entire human transcriptome from GENCODE GRCh38, which consists of mature RNA sequences. To preprocess the sequences, we first removed any duplicates to ensure data integrity. The remaining sequences were then tokenized into overlapping 6-mer tokens, resulting in a maximum input length of 510 tokens per sequence to fit in the 512 token maximum after appending [CLS] and [SEP] tokens. Sequences that resulted in more than 510 6-mer tokens were split into separate inputs without overlaps, since motifs are presumably already captured by the overlapping 6-mer tokens. Hence, this tokenization strategy was chosen to capture local sequence dependencies effectively and potentially learn regulatory motifs in mRNA sequences.

We made slight modifications to the default BERT architecture to better suit the characteristics and data size of mRNA sequences. Specifically, mRNAbert was configured with 6 Transformer layers and 6 attention heads per layer. The model was trained using mixed-precision floating-point arithmetic to optimize performance and memory usage.

Training was conducted on 4 NVIDIA Tesla P100 GPUs over a period of 3 weeks. The training process involved the following hyperparameters: a learning rate of 4e-4, an effective batch size of 250, and a total of 200,000 training steps. During training, we employed techniques such as gradient accumulation and learning rate scheduling to enhance model convergence and stability.

### rG4 formation and subtype classification datasets

We developed G4mer for rG4 formation prediction by fine-tuning mRNAbert on the binary rG4 formation dataset. The preparation of the training dataset includes extracting 5528 experimentally detected rG4 sequences based on the rG4-seq[1] experimental protocol that have been post-processed and published as the rG4-seeker dataset[34]. Duplicated sequences from the same genes that were found by different RT stops were dropped. This left us with 5454 unique rG4 sequences. Previous work on DNA G4 showed that flanking regions play an important role in the formation and stability of the structure[23]. As such, the rG4 sequences are mapped to the GRCh38.v29 transcriptome sequences to obtain their flanking regions to the left and right of the putative rG4, forming 5438 unique sequences with a maximum length of 70 nt. Some extended sequences were duplicated when an rG4 is a subset of another rG4, resulting in the same sequence after extension. rG4 sequences that couldn't be mapped to a transcript were stored as their original sequences without additional flanks.

A similar number of 5672 negative sequences were obtained by querying all highly expressed genes in the HeLa cell line with no experimentally found rG4s, and looking for a relaxed putative G4 regular expression G(2+)-N(1-30)3-G(2+). The sequences are set to be the same length as the positive sequences, with the matched pattern centered in the sequences. However, since there is a slight overrepresentation of long patterns found, we set a limit for the number of long patterns included in the training data. We did this by first setting a limit of 1200 sequences sampled for each bin of 10 nt in the range of 10 nt to 60 nt. The combination of both positive and negative sequences resulted in a total of 11,110 sequences used in the binary dataset.

To develop G4mer for multiclass classification, mRNAbert was fine-tuned on a multiclass dataset. The 5454 unique experimentally detected rG4 sequences from rG4-seeker were classified into eight classes: canonical/G3L1-7, bulges, longloop, two-quartet, G ≥ 40%, potential G-quadruplex & G ≥ 40%, potential G-triplex & G ≥ 40%, and unknown. These sequences were categorized into the eight subclasses based on sequence motifs, loop lengths, and guanine content, following the hierarchical assignment framework described in the Methods section of Kwok et al. (2016)[1] and further refined using the rG4-seeker pipeline for classification[34]. Finally, sequences that did not fit into any of the above structural categories were assigned to the Unknown category. When a sequence matched multiple classifications,

it was assigned to the highest predicted stability class, following the hierarchical ranking of Canonical rG4s > Long loops > Bulges > Two-quartet rG4s, and so on. We obtained the transcriptomic flanking regions for each of the multiclass-labeled sequences when possible to similarly obtain the desired length as for the binary dataset. This resulted in a fine-tuned G4mer model for multiclass classification.

## G4mer training and evaluation

For evaluation, we implemented a 10-fold CV strategy to compare the performance of transformer-based G4mer and CNN-based G4Detector. Both models were trained and tested on the same binary and multiclass datasets. For the binary classification CV, G4mer was configured with a learning rate of 2e-4, a batch size of 32, and fine-tuned for 2 epochs. G4Detector was run with its default hyperparameters: 256 filters, a kernel size of 12, a batch size of 128, a learning rate of 1e-3, and a hidden size of 32, and was trained for 1 epoch. For the multiclass CV, the hyperparameters for both models remained the same as in the binary classification. However, we implemented early stopping with patience of 5 for both models, training for a maximum of 50 epochs. The models with the best validation loss were selected for validation. The primary metrics used to assess performance were accuracy and the area under the receiver operating characteristic curve (AUC) for both binary and multiclass classification tasks. The results from all ten folds were averaged to obtain the final performance metrics, providing a robust evaluation of the models' predictive capabilities.

Using the optimal hyperparameters derived from the CV, G4mer was developed for both binary and multiclass classification. For binary classification, G4mer was trained with a learning rate of 2e-4, a batch size of 32, and trained for 2 epochs using the binary dataset. For multiclass classification, G4mer was trained with the same set of hyperparameters for 5 epochs using the multiclass dataset.

To evaluate the specificity of G4mer's predictions in a population-wide context, we estimated the false discovery rate (FDR) using highly expressed genes in HeLa cells. Specifically, we identified transcripts with no detected rG4 signal in rG4-seq experiments and treated regions within these transcripts as likely true negatives. We then applied G4mer to score these regions and considered predictions with a G4mer probability score > 0.7 as false positives. The FDR was calculated as the fraction of these high-scoring predictions among all predictions made in the rG4-negative background.

## Model evaluation on G4RNA database

To compare how G4mer performs on sequences from other experimental protocols of various lengths, we extracted sequences from G4RNA. G4RNA sequences were obtained from the G4RNA database[35]. In the data preprocessing phase, duplicate entries were removed from the dataset if they had the same sequence, experimental protocol, and rG4 formation results, eliminating any redundancy that could potentially skew the results of the analyses. Sequences that have disagreeing labels across different experimental protocols, have NaN labels, or appear in the training dataset of G4mer were excluded. The pre-processing step resulted in 795 sequences of lengths ranging from 14 nt to 1368 nt, each validated by one of 24 experimental protocols for rG4 formation.

The final set of G4RNA sequences was used to validate G4mer's performance. Additionally, we ran three other models—cGcC, G4Hunter, and rG4detector—on the same set of sequences. rG4 predictions of G4RNA sequences of cGcC[18] and G4Hunter[19] were obtained from G4RNA Screener webtool[70] that runs both methods in parallel. The default hyperparameters were used, such as the window size of 60 for cGcC and score thresholds of 4.5 for cGcC and 0.9 for G4Hunter. The maximum prediction score per sequence was assigned as the final score. Predictions of rG4detector (https://github.com/OrensteinLab/rG4detector) were obtained by running the tool locally.

## Model interpretation with EIG for flanking sequence importance

To investigate the contribution of rG4 flanking regions to rG4 predictions made by the G4mer model, we utilized EIG for model interpretation. EIG quantifies the importance of individual input features by integrating the gradients of the model's output with respect to its input, following a path from a baseline class to the input sample (class of interest)[37]. In this context, the baseline serves as a counterfactual reference, reflecting a condition where no rG4 predictive signal is present, while the input sample represents an rG4-containing sequence. By computing and aggregating gradients along a linear or non-linear path between the baseline and the input, EIG provides a detailed attribution score for each feature. For this analysis, our baseline points are samples taken from the set of non-rG4 sequences obtained from G4mer's training data, which serve as the starting point for gradient integration. Specifically, following[37], we employed three integration paths for improved robustness, each starting from one of three non-rG4 sequences whose representations are closest to the median representation of all sequences in the baseline class. From each of those reference points, we then compute EIG values for the set of sequences predicted to contain canonical rG4 sequences (G4mer score > 0.7) with the rG4 motif centered as much as possible. For each path, we used 200 integration steps, and for each sample, we averaged the individual paths from the three reference points described above. Since the inputs to G4mer are 6-mers, we obtained an attribution score for every 6-mer input through EIG. We zeroed out the attribution scores for 6-mers that overlapped with the rG4 motifs to focus on flanking regions. We then identified the 6-mers in the rG4 flanking regions that showed significant contributions to the model's predictions. To assess the significance of these attributions, we followed the procedure in[37] and conducted a two-sided t-test comparing the k-mer attribution scores of each token and a random set of tokens, adjusting for multiple comparisons using Benjamini-Hochberg procedure with a significance threshold of FDR < 0.05 to determine statistical significance. After obtaining significant 6-mer tokens, we then assigned attribution scores for each nucleotide, as well as 2-mers and 3-mers, by using the maximum attribution score of the significant 6-mer tokens in which they reside. To ensure we captured the relevant flanking regions surrounding the rG4 motifs, we extended our analysis to include 20 nucleotides upstream and downstream of the predicted rG4s.

## Perturbation analysis of rG4 flanking regions

To evaluate the contribution of individual nucleotides in the flanking regions of predicted rG4s, we performed a perturbation analysis by systematically mutating guanine (G) and cytosine (C) residues. We selected high-confidence rG4-forming sequences (wild-type G4mer score > 0.7) containing a canonical rG4 motif and identified up to 60-nucleotide flanking regions on both sides of the motif. Within these flanks, we generated in silico mutations by substituting each G with a C and each C with a G one at a time, creating a mutated version of the full sequence for each perturbation.

We then computed the change in G4mer score (ΔG4mer) as the difference between the mutated and wild-type predictions. The average ΔG4mer values were aggregated by mutation type and position relative to the canonical motif to assess directional effects on rG4 probability. This analysis allowed us to quantify the stabilizing and destabilizing effects of nucleotide substitutions in the flanking regions of high-confidence rG4-forming sequences.

## RBP motif enrichment analysis

To explore whether rG4 flanking sequences are enriched for RBP motifs, we analyzed the UTR sequences surrounding canonical rG4 motifs within predicted rG4 regions, with the motifs defined by the regular expression G{3+}N{1,7}G{3+}N{1,7}G{3+}N{1,7}G{3+}, where N represents any nucleotide (A, C, G, or U). We separated the sequences into two groups: (1) those predicted by G4mer to form rG4 structures

(G4mer score > 0.7), and (2) those not predicted to form rG4s (G4mer score < 0.3). For each sequence, we located the canonical motif and extracted its flanking regions (excluding the motif itself), generating matched sets of forming and non-forming flanks. Motif scanning was performed on the forming and non-forming sets of flanking sequences using FIMO from the MEME suite[39] with default settings on the CISBP-RNA motif database[38], converted to MEME format.

For each RBP motif, we computed the number of unique flanking sequences in which the motif was detected in both forming and non-forming groups. A $\log_2$ fold enrichment score was calculated as:

$$\log_2 \left( \frac{\text{fraction of forming sequences with motif} + \varepsilon}{\text{fraction of nonforming sequences with motif} + \varepsilon} \right) \quad (1)$$

where $\varepsilon = 1 \times 10^{-6}$ was added for numerical stability. Only those with sufficient hits (at least five motif hits) across groups were retained. Statistical significance was assessed using one-sided Fisher's exact test. We applied Benjamini-Hochberg FDR correction to the p-values, and only motifs with FDR-adjusted $q < 0.05$ and $\log_2$ fold enrichment > 1 were reported in the plot.

To visualize representative RBP motifs enriched in the flanking regions of predicted rG4 regions, we generated sequence logos from published position weight matrices (PWMs). PWMs for HNRNPF and ELAVL1 were obtained from the CISBP-RNA[38] database (http://cisbp-rna.ccbr.utoronto.ca), while the SRSF4 motif PWM was retrieved from the ATtRACT[78] database (https://attract.cnic.es). All PWMs were standardized as position probability matrices and plotted using the Logomaker Python package (v0.8). Logos were plotted using matplotlib and saved as high-resolution vector PDFs for inclusion in the main figures.

### gnomAD variant analyses

For the analysis of gnomAD variants in UTR regions, we utilized the gnomAD v3.1.2 dataset (https://gnomad.broadinstitute.org/downloads#) for all autosomal chromosomes. Variants were filtered to retain only those that passed all quality control (QC) filters, as indicated by having a FILTER value of None. This approach ensures that only high-confidence variants are included in the analysis, reducing the potential impact of sequencing artifacts or low-quality variant calls. We further excluded variants located in low complexity regions, decoy regions, and segmental duplications by checking for the presence of 'lcr', 'decoy', or 'segdup' in the INFO field. These regions are known to introduce biases and inaccuracies in variant calling and are therefore excluded to enhance the reliability of our results. Next, to ensure variants came from reads with high sequencing depth in the gnomAD WGS dataset, only those with a total observed allele count of at least 80% of the maximum number of sequenced alleles (152,312) were retained.

Since transcripts can exhibit varying levels of constraint and selection pressure, we assigned genomic constraints from the gnomAD database (https://gnomad.broadinstitute.org/downloads) to the filtered variants. This allowed us to compare the allele frequency of variants with similar levels of genomic constraints. To achieve this, one group of variants was divided into quartiles based on their genomic constraint scores. For each quartile, we randomly sampled variants from the other group. The stratified sampling ensured that both groups contained variants that had closely matched quartile distributions.

To analyze rG4-breaking variants using the CADD metric, we retrieved raw CADD scores for each variant from the gnomAD Hail Table, which assigns a raw CADD score for every variant based on its predicted deleteriousness. Specifically, we intersected our list of rG4-breaking variants with the gnomAD data, matching each variant by its locus, reference allele, and alternate allele. The matched raw CADD scores reflect the relative likelihood that a given variant is deleterious

based on a wide range of annotations, including sequence conservation and regulatory impact, where a positive CADD score represents increased potential deleteriousness. The scores were then used in the analyses to assess the potential functional impact of variants that disrupt rG4 structures. For the rG4-breaking variant in 5′ UTR, we obtained the raw CADD scores for all variants. For 3′ UTR, we focused on rG4-breaking variants in disease genes, particularly those with LOEUF upper bound values < 0.9 from the gnomAD v4.1 transcript constraint file.

### ClinVar variant analysis

We downloaded the ClinVar GRCh38 VCF release from March 2025, obtained via the NCBI FTP site (ftp://ncbi.nlm.nih.gov/pub/clinvar/vcf_GRCh38/, and filtered for SNVs located in annotated 5′ and 3′ UTRs. We focused on variants with clear clinical such as Pathogenic and Benign. For each variant, we computed the G4mer prediction score for both the wild-type and mutant transcript sequences and retained only those located in putative rG4 regions (G4mer score > 0.7). When a variant mapped to multiple transcripts, we selected the instance with the minimum ΔG4mer score (defined as mutant score - wild-type score), ensuring that each variant was represented only once to avoid skewing the distribution. A one-sided Mann-Whitney U test was performed to compare the distributions of ΔG4mer values between the pathogenic and benign groups.

### Penn medicine bioBank

Penn Medicine BioBank (PMBB) hosts a comprehensive dataset of seven years of longitudinal electronic health records (EHR), genetic sequencing, and biological samples of study participants. The PMBB study's dataset encompasses a subgroup of over 44,000 individuals who underwent Whole Exome Sequencing (WES). DNA was isolated from the stored buffy coats of these individuals, and exome sequencing was carried out by the Regeneron Genetics Center in Tarrytown, NY, aligning the sequences with the GRCh38 reference genome as previously outlined[32]. In preparation for further phenotype analysis, samples were excluded based on criteria such as low exome sequencing coverage (below 85% of targets reaching 20x coverage), high rates of heterozygosity/contamination (D-stat > 0.4), genetically identified sample duplicates, and discrepancies between reported and genetically verified sex.

In this study, we focused on samples from European and African ancestries, which together represent the largest genotyped groups in PMBB. Among genotyped individuals, 66.8% are of European ancestry, and 24.6% are of African ancestry[32], enabling well-powered analyses. We further filtered samples and variants in PMBB to be included in our analyses. We restricted the analyses to only keep samples that are 2nd-degree unrelated in each ancestry group, resulting in two groups of samples of European (N = 29,362) and African (N = 10,217) genetic ancestries. Furthermore, exclusion criteria applied to variants include singletons and high missing call rates (exceeding 0.1).

To obtain PMBB variants that are rG4-altering, we mapped variants across all transcripts and obtained the wild-type regions around the variants. rG4-breaking variants were defined by a wild-type G4mer score above the rG4 threshold of 0.5, with the mutated sequence score at least 0.2 lower than the wild-type to indicate a reduction in rG4 structure stability in the presence of the variant. Conversely, rG4-forming variants were characterized by a wild-type score below 0.5, indicating the absence or weak formation of rG4, and a mutated score at least 0.2 higher, suggesting increased rG4 probability or formation in the presence of the variant.

### Phenome-wide association studies

We tested the association of selected PMBB rG4-altering variants in breast cancer-associated genes with phenotypes extracted from PMBB[32]. ICD-10 codes from PMBB samples were mapped to distinct

disease entities (i.e., phecodes) via Phecode Map 1.2 using the PheWAS package in R[79]. To establish a patient as a 'case' for a given phecode, we required a minimum of 2 counts of the code, as repeated diagnoses of a code on different days improve phenotype precision[80]. Our association analyses considered only phecodes with at least 20 cases, based on prior simulation studies for power analysis of common and rare variant gene burden PheWAS on binary traits, which highlights the impact of case counts on the power to detect genetic associations[81,82]. This criterion led to the inclusion of 957 phecodes.

Each phecode was tested for association with each rG4-altering variant using a logistic regression model adjusted for sex, age at enrollment, age-squared, and the first 10 principal components of genetic ancestry. For each variant, ancestry-specific PheWAS was first performed for two major groups: African (AFR) and European (EUR) ancestry. To account for population structure, we performed ancestry-stratified PheWAS in individuals of African (AFR) and European (EUR) ancestry, followed by meta-analysis. While principal component adjustment corrects for broad-scale population structure, ancestry-specific genetic differences in allele frequencies and linkage disequilibrium patterns may persist and influence association signals[83,84]. Stratified analyses allow for the detection of ancestry-specific associations and mitigate potential biases introduced by differences in genetic architecture between populations[83,85]. The cross-ancestry summary results were then meta-analyzed using an inverse-variance weighted random-effects model[32] to combine evidence across populations while accounting for potential heterogeneity in effect sizes. The odds ratio (OR) from the meta-analysis of combined ancestry is used to determine the direction of each arrowhead in the association plots, where OR > 1 suggests an increased risk with an up arrowhead. While stratified PheWAS followed by meta-analysis allows us to account for ancestry-specific effects, some variants were observed in only one ancestry group. For instance, the rG4-disrupting variant in *EPN3* was exclusively present in individuals of European ancestry, hence the stratified and meta-analyzed result remained the same.

To address multiple testing, an association between variant and phecode with FDR p < 0.1 was considered significant. Thus, the adjusted threshold for significance was p << 5.53e-2. In addition, we calculated a more stringent Bonferroni corrected threshold for each gene of p << 9.78e-5. We then filtered the rG4-altering variants for functional validation using dual luciferase assays by considering those with associations that pass the Bonferroni-adjusted significance threshold.

## Plasmid preparation, cell culture, and transfections

In order to perform a bicistronic dual luciferase assay to quantify changes in expression due to variants in predicted rG4s, we created a modified version of pMiRcheck2[86], puORF-Check3 (Supplementary Fig. 6). We obtained the WT 5′ UTR sequences of *EPN3* and *MSH6* (387 and 188 nt, respectively), and the first 12 nt of CDS, to clone upstream of the stuffer sequence in the puORF-Check3 plasmid backbone using the NdeI restriction site (NEB HiFi). Mutant sequences of each 5′ UTR were cloned in the same manner as well. All constructs were verified by whole plasmid sequencing. HuH-7 cells (JCRB, Cat# JCRB0403), CHO-K1 cells (ATCC, Cat# CCL-61), and NIH3T3 cells (ATCC, Cat# CRL-1658) were used for conditional expression of reporter genes. For transient transfections, cells were seeded 1 day before transfection in 24-well plates at a density of 80,000 cells per well. 60 ng of the dual luciferase plasmid was transfected into each well using Lipofectamine 3000 following the manufacturer's protocol, with 3-4 technical repeats for each construct. 3-5 biological replicates were obtained by transfecting cells from separate passages on separate days using newly prepared reagents. HuH-7 and NIH3T3 cells were cultured in Dulbecco's modified Eagle's medium (DMEM), and CHO cells were cultured in F12 medium, both supplemented with 10% fetal bovine serum.

## Dual luciferase reporter assays

Luminescence was measured using the Promega Dual-Luciferase Reporter Assay System (E1910) following the manufacturer's protocol. Cells were lysed by adding 100 μL of lysis buffer, and 20 μL of each lysate was transferred to a white opaque 96-well plate. With Renilla luciferase serving as the reporter and Firefly luciferase as the internal control, the intensities of Firefly and Renilla luciferase luminescence were measured with a microplate reader (BioTek Synergy Neo2 multimode reader) after automatic injection (BioTek Dual Reagent Injector by Agilent) of the Luciferase Assay Reagent II and Stop & Glo reagents. For each test construct, the Firefly and Renilla luciferase activities were inferred from the measured counts per minute to obtain the relative Firefly-to-Renilla expression ratio. Measurements for mutated constructs were then normalized to the wild-type construct to provide a fold-change measure for the translational efficiency conferred by each variant[87]. Statistical significance was determined using an independent t-test, comparing the relative Firefly-to-Renilla expression ratios across transfections of each wild-type and mutated construct pair for all cell lines.

## Circular dichroism (CD) spectra

RNAs were purchased from Integrated DNA Technologies (IDT) and prepared as follows: RNA stock solutions were diluted to a final concentration of 40 μg/mL in a buffer containing 150 mM KCl or 150 mM LiCl. Prior to analysis, all RNA samples were heated to 100℃ for 3 minutes and then allowed to cool to room temperature.

Circular dichroism (CD) spectra were collected using a Chirascan V100 spectrometer over a wavelength range of 220–300 nm, with a step size of 0.5 nm, a bandwidth of 1 nm, and an integration time of 1.25 seconds per data point. Spectra were recorded at 25℃ for initial measurements. Following the initial spectra collection, the sample chamber temperature was increased to 95℃ using a Peltier controller, and the samples were allowed to stabilize for 3 minutes before additional spectra were acquired. Baseline spectra of the buffer (150 mM KCl or 150 mM LiCl) were subtracted from the RNA spectra using the Chirascan software to ensure accurate measurements. All CD spectra were obtained in three independent replicates under each condition.

## Reporting summary

Further information on research design is available in the Nature Portfolio Reporting Summary linked to this article.

## Data availability

In this work, we used rG4-seeker-processed data[34] to develop G4mer, derived from publicly available raw data generated by the rG4-seq protocol on RNA from human HeLa cells[1] (GEO accession code GSE77282). The rG4-seeker dataset used to construct the rG4 binary and multiclass training sets is available as supplementary material in Chow et al. (2020). Benchmarking sequences from G4RNA were downloaded from the G4RNA database (http://scottgroup.med.usherbrooke.ca/G4RNA/). Transcript sequences and annotation files for the human transcriptome (GRCh38, v29) were obtained from GENCODE. Population allele frequencies, genomic constraint scores, and variant annotations were retrieved from gnomAD v3.1.2 and v4.1 (https://gnomad.broadinstitute.org/downloads#), and raw CADD scores were accessed from the CADD database (https://cadd.gs.washington.edu/). ClinVar variant annotations (GRCh38, March 2025 release) were obtained from the (NCBI FTP site) and used to assess rG4-breaking effects in clinically annotated UTR variants (see Supplementary Data 1). RBP motif position weight matrices (PWMs) were obtained from the CISBP-RNA and ATtRACT (https://attract.cnic.es/) databases. ICD-10 codes in PMBB were mapped to phecodes using Phecode Map 1.2 from the PheWAS R package (https://phewascatalog.org/phecodes). Individual-level phenotype and genotype data from the Penn Medicine BioBank (PMBB) are subject to institutional data use agreements and patient privacy regulations, and access is limited to

qualified researchers via the PMBB application process (https://pmbb.med.upenn.edu/investigators.php). Sequences used in dual luciferase experiments and circular dichroism experiments are available in Supplementary Tables 2 and 3, respectively. All data generated in this study are available in the Bitbucket repository (https://bitbucket.org/biociphers/g4mer/src/main/) and archived on Zenodo (https://doi.org/10.5281/zenodo.16912798)[88].

## Code availability

All scripts used in this analysis, including code for fine-tuning and evaluating the G4mer model, are available at the following Bitbucket repository: https://bitbucket.org/biociphers/g4mer/src/main/ (archived at Zenodo, https://doi.org/10.5281/zenodo.16912798[88]). Pre-trained models, including mRNAbert, G4mer (binary), and G4mer-subtype (multiclass), are hosted on HuggingFace (https://huggingface.co/Biociphers/mRNAbert, https://huggingface.co/Biociphers/g4mer, https://huggingface.co/Biociphers/g4mer-subtype). Access requires users to log in and agree to an academic usage license, which permits internal, non-commercial use by individuals at academic or not-for-profit institutions. A web application to run G4mer predictions is available at https://tools.biociphers.org/g4mer/index. Use of the webtool is also governed by the academic usage license. For commercial licensing inquiries, please contact the corresponding author.

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

## Acknowledgements

We sincerely thank Matthew Gazzara, Seong Woo Han, Kevin Yang, David Wang, and Di Wu for their contributions and suggestions. Our appreciation extends to the members of the Barash lab for discussions and thoughtful feedback. Special thanks to Michelle Scott and Jean-Michel Garant (Sherbrooke University) for sharing sequences in the G4RNA database. Finally, we gratefully acknowledge the staff of the Regeneron Genetics Center for whole-exome sequencing of PMBB participants. This work was supported by NIGMS grants T32GM148376 (A.J.); R35GM142864 (D.D.); R25GM055366 (B.G.); NHGRI grant T32HG009495 (D.G.); The Zuckerman-CHE STEM Leadership Program (D.G.); CureBRCA and the Basser Center for BRCA pilot grants (Y.B., K.N.); NIH grants R01 LM013437 (Y.B.); R01 GM-147739 (Y.B., N.H.); and NSF Cooperative Agreement DBI-2400327 (Y.B.).

## Author contributions

F.Z. and Y.B. conceived and designed the project. F.Z. developed mRNAbert and G4mer, performed model performance analyses, variant analyses, and disease association studies with ancestry-based PMBB variants under the guidance of Y.B. PMBB samples and variants for disease association analyses filtering were done by F.Z. and D.G under the guidance of Y.B. Selection of disease-associated variants and sequences for validation were done by F.Z. and D.G under the guidance of Y.B. and K.N. N.I. and F.Z. performed EIG analyses under the guidance of Y.B. N.H designed puORF-Check3. D.G. planned and performed the cloning and dual luciferase experiments under the guidance of Y.B. and N.H. Dual luciferase result analyses were done by D.G. and F.Z. under the guidance of Y.B. F.Z. and D.G. selected sequences for structural validations under the guidance of Y.B. and D.D. A.J. and B.G. performed circular dichroism experiments and analyses under the guidance of D.D. D.G. provided the methods for dual luciferase experiments. A.J. provided methods for circular dichroism experiments. F.Z. and S.J. designed the web tool, and S.J. built the web tool under the guidance of Y.B. F.Z. wrote the paper, and all authors contributed to editing the paper.

## Competing interests

The authors declare no competing interests.
