## [Transparent Peer Review file · Nature Communications]

G4mer: An RNA language model for transcriptome-wide identification of G-quadruplexes and disease variants from population-scale genetic data

Corresponding Author: Professor Yoseph Barash

Version 0:

Reviewer comments:

Reviewer #1

(Remarks to the Author)

The authors introduce “G4mer”, an RNA language model for predicting RNA G-quadruplexes (rG4s). G4mer outperforms existing models and exhibits greater resilience to variations in input sequence lengths. G4mer allowed the authors to investigate the transcriptome-wide effects of genetic variants on rG4 formation. Variants predicted by G4mer to significantly disrupt rG4s were found to be under negative selection. They found that rG4-altering variants in the disease-associated EPN3 and MSH6 genes may be associated with diseases. Finally, they discuss the limitations of their approach.

Overall, this is an interesting article that deserves publication pending the following minor changes:

- The authors report a heightened negative selection and increased vulnerability to genetic variants for longer rG4s, arguing “that their structures are weaker and more flexible, allowing a single nucleotide variation to strongly affect their structure or binding by RNA-binding proteins (RBPs) that target these regions”. This is something I do not understand – it seems at first glance that it should be easier to keep G4 forming potential within a long sequence, especially when multiple G-runs are present, acting as “spare tyres” at the RNA level. Could the authors comment on that?
- Fig 3B : should be easy to add the G4 candidate sequence- not simply the SNP.
- Fig 4B: I am surprised by the near 0 / flat line CD spectra of the non-G4 forming sequence (dark blue) at 25°C: single stranded nucleic acids generally give a signal at room temperature due to base stacking – a flat line is generally observed only at high temperature.

(Remarks on code availability)

Reviewer #2

(Remarks to the Author)

The authors introduce G4mer, an RNA language model based on the transformer architecture, designed to predict RNA G-quadruplexes (rG4s) and assess the effects of genetic variants on rG4 formation. The study integrates computational modeling, experimental validation, and disease association analysis to explore rG4s' functional and structural roles. The authors claim that G4mer outperforms existing tools, such as CNN-based G4detector, scoring-based cGcC, and G4Hunter, particularly on longer transcriptomic sequences and datasets from diverse experimental protocols. The model was further applied to study the functional significance of rG4-altering variants in diseases, including breast cancer, using phenome-wide association studies (PheWAS) and experimental approaches like dual luciferase reporter assays and circular dichroism spectroscopy. I have the following major concerns regarding the work.

1. In the initial experiments involving binary classification tasks with rG4-seekers, although G4mer outperforms G4Detector, both achieve high AUC scores exceeding 0.9. This suggests that detecting or distinguishing rG4 from other structures may be relatively straightforward. However, when transitioning to subtype classification, G4mer demonstrates significantly superior performance compared to G4Detector. If the finer-grained classification is more significant, the subsequent analysis

pertaining to subtypes seems to receive less attention. It appears that the binary classification version suffices for the ensuing experiments.

2. Although the EIG analysis provides insights into the importance of flanking sequences, other interpretable features, such as specific sequence motifs or secondary structures, are not deeply explored.

3. On the other hand, EIG is not suitable for analyzing large-scale language models due to significant deviations in the integrated gradients of the input. In my view, the attributions from this method may not be sufficiently appropriate to draw the conclusion that "sequences flanking the G-tracts do indeed contribute to the rG4 predictions." I suggest utilizing a perturbation-based method directly to obtain input attributions, similar to the approach you employed in predicting the rG4-altering effects of single nucleotide variants. You can alter the nucleotides in the flanking region, and observe the changes of the prediction.

4. It appears that your experiments concerning rG4 alterations primarily focus on single nucleotide variants. Have you considered the impact of multiple nucleotide variants? In the section analyzing PMBB variants, do they involve alterations of multiple nucleotides? Perhaps it would be beneficial to delve deeper into discussions and analyses on natural mutations rather than artificial negative selections. Understanding the relationships between nucleotides is crucial and prevalent in nature.

5. While the study validates rG4-altering variants in two genes (EPN3 and MSH6) for breast cancer, it does not extend this approach to other disease-associated genes or phenotypes. A broader experimental validation across multiple variants would strengthen the claims.

6. The model is trained exclusively on human transcriptomic data. It is unclear whether G4mer could generalize to predict rG4s in other organisms, such as plants or bacteria, where rG4s also play regulatory roles.

7. While G4mer achieves high-performance metrics, the reliance on a single experimental dataset (rG4-seq) raises concerns about overfitting to specific experimental artifacts. The authors should have included independent validation datasets from additional sources, such as SHAPE-MaP or other RNA structure-seq methods.

8. Lack of clinical validation metrics: The study applies G4mer to predict rG4-breaking variants in disease-associated genes, but the clinical implications of these predictions are not quantified. For example, how well does G4mer distinguish between pathogenic and benign variants in ClinVar? Are there false positives in the predicted rG4-breaking variants from gnomAD?

9. While the authors emphasize the importance of longer rG4s, the biological mechanisms underlying their functional significance remain speculative. Experimental validation to confirm whether longer rG4s are more prone to disruption by variants would add value.

10. In terms of the structural validation of the two specific rG4 variants from EPN3 and MSH6, providing the specific secondary structure would enhance clarity and intuition. Additionally, the manuscript lacks any structural visualization of these variants, which diminishes the overall intuitiveness of the work.

11. I got "Repository not found" when opening the code link.

Typos:

page9, " $\leq 28\text{nt}$ " seems to be " $\geq 28\text{nt}$ "

(Remarks on code availability)

I got "Repository not found" when opening the above link. The code is unavailable during my reviewing period.

Reviewer #3

(Remarks to the Author)

Review Nature Communications

Thank you for allowing me to review this paper. It is very strong but with a few caveats for improvement. Please see below for my comments.

"We first evaluated its performance against current state-of-the-art methods across diverse data and experimental techniques, showing that it significantly outperforms existing methods"

Comment: Diverse data like what? Do you mean the makeup of the individuals in the dataset or just a collection of different datasets? This might be misleading language.

"On this task, G4mer outperformed G4Detector, obtaining an accuracy of 0.95 and a receiver operating characteristic area under the curve (ROC-AUC) score of 0.98, compared to an accuracy of 0.92 and an AUC of 0.97 for G4Detector (Fig.1c)." And related metrics throughout.

Comment: Accuracy is a notoriously problematic measurement for model validation in classification- You include ROC here so you've covered sensitivity (aka recall) and specificity, but adding precision-recall AUC I think makes a stronger argument instead of just ROC and accuracy. You've already calculated this in the supplement so it should be an easy add.

"In this data, the authors categorized each sequence into one of eight labels, ranging from canonical to known noncanonical subtypes such as long loops, bulges, and two-quartets."

Comment: Is this a standard categorization technique or was this using content area expertise to develop the classification? Be more specific about how this classification was done.

Comment: There is good detail here regarding the training and evaluation of the model

"We further filtered samples and variants in PMBB to be included in our analyses. We restricted the analyses to only keep samples which are 2nd-degree unrelated in each ancestry group, resulting in two groups of samples of European (N=29,362) and African (N=10,217) genetic ancestries. Furthermore, exclusion criteria applied to variants include singletons and high missing call rates (exceeding 0.1)."

Comment: It would be beneficial for the authors to describe how they might be able to use this methodology in other populations besides European and African Descent.

"ICD-10 codes from PMBB samples were mapped to distinct disease entities (i.e. phecodes) via Phecode Map 1.2 using the PheWAS package in R71."

Comment: Spelling error

"Each phecode was tested for association with each rG4-altering variant using a logistic regression model adjusted for 24 sex, age at enrollment, age-squared, and the first 10 principal components of genetic ancestry. For each variant, ancestry-specific PheWAS was first performed for two major groups: African (AFR) and European (EUR) ancestry."

I don't understand why you've stratified by genetic ancestry and then recombine using a meta analysis when you've already controlled for genetic ancestry in the regression via principal components. Your beta estimates should already be adjusted in the first stage without stratifying. Provide more justification for doing the strata and why you meta analyzed or else just go with the non stratified phewas on the whole group. If you are going to stratify and recombine, it would be helpful to have the individual phewas results available in the supplement as well.

Overall this is a very good paper but needs clarity on the methods used in phewas. Additionally, many genetics methodologies are validated on non-diverse dataset and have the potential to perpetuate biases. The authors should spend time discussing how one might validate this method further in genetic ancestries besides European and African.

(Remarks on code availability)

I am very pleased with how the authors organized everything in their GitHub repo. The file structure is very intuitive and they provide complete Jupyter notebooks to reproduce their results.

Version 1:

Reviewer comments:

Reviewer #1

(Remarks to the Author)

The authors have addressed all my comments in a satisfactory manner. I recommend acceptance.

(Remarks on code availability)

Reviewer #2

(Remarks to the Author)

Thank you for your response. However, my previous concerns have not been completely resolved.

For point 1:

There are still some questions regarding the rebuttal's analyses. The authors state that 'Two-quartet and bulge subtypes tended to show more severe decreases in G4mer scores upon mutation'. I can identify the Two-quartet subtypes from the presented figure, but the Bulges in the 5' UTR panel still exhibit very high values. Additionally, why is the G-quadruplex not sensitive to the SNV? Considering its name definition requiring four Gs in its structure, one would expect it to be more sensitive? Moreover, I am curious whether G4mer is reliable in reflecting the changes brought about by the SNV.

For point 3:

Thank you for incorporating alternative interpretation methods to cross-validate with your EIG; I believe these additions will fortify your findings. One suggestion is to delve deeper into the discrepancies between these two methods.

For point 10:

I concur with your insights on the current structure prediction tools. However, have you considered exploring secondary structure prediction tools? It might offer additional perspectives for prediction comparison. Inviting a visitation of the structure could enhance the paper's presentation.

Minor:

Why is there no median line in the box plot of the purple box in Figure 3b?

(Remarks on code availability)

The readme is too simple to set up the environment for me. It's hard for me to run the entire code.

Reviewer #3

(Remarks to the Author)

The authors did a wonderful job addressing my comments and satisfactorily addressed each of my concerns. No further comments from me.

Code link in bitbucket needs to be addressed though.

(Remarks on code availability)

The direct link above says "something went wrong" but if you dig into the code base more, you can see the commits etc. The authors need to update this.

Version 2:

Reviewer comments:

Reviewer #2

(Remarks to the Author)

Thank you for the revision! It looks nice to me now. I would suggest to make the code into GitHub, which is easier to access.

(Remarks on code availability)

It's better to make it into GitHub, which is easier to access.

made.

We thank all the reviewers for their careful evaluation of our work and their thoughtful suggestions. In response, we have carefully edited the manuscript and added additional analyses, which are reflected in a new main figure and several supplementary figures. A point by point response to reviewers' comments is attached below. We believe the resulting edits and additions have contributed to significant improvements of the manuscript. We would like to thank again the reviewers for putting in the time and effort to help us improve our work, and we hope you will find the work suitable for publication.

Reviewer #1 (Remarks to the Author):

The authors introduce “G4mer”, an RNA language model for predicting RNA G-quadruplexes (rG4s). G4mer outperforms existing models and exhibits greater resilience to variations in input sequence lengths. G4mer allowed the authors to investigate the transcriptome-wide effects of genetic variants on rG4 formation. Variants predicted by G4mer to significantly disrupt rG4s were found to be under negative selection. They found that rG4-altering variants in the disease-associated EPN3 and MSH6 genes may be associated with diseases. Finally, they discuss the limitations of their approach.

Overall, this is an interesting article that deserves publication pending the following minor changes:

We sincerely appreciate the reviewer's thoughtful evaluation of our work and their recognition of G4mer's contributions to RNA G-quadruplex (rG4) prediction and variant analysis. We are grateful for the constructive feedback and have carefully addressed each of the suggested changes below.

- The authors report a heightened negative selection and increased vulnerability to genetic variants for longer rG4s, arguing “that their structures are weaker and more flexible, allowing a single nucleotide variation to strongly affect their structure or binding by RNA-binding proteins (RBPs) that target these regions”. This is something I do not understand – it seems at first glance that it should be easier to keep G4 forming potential within a long sequence, especially when multiple G-runs are present, acting as “spare tyres” at the RNA level. Could the authors comment on that?

The reviewer raises a good question. Previous work using biochemical assays have demonstrated that G-run length and intervening loop length directly influence rG4 stability (e.g., PMID: 21744844 PMID: 23683360). A recent high-throughput study from the Dominguez lab clearly shows that lengthening the loop generally weakens rG4, but the effects are context-specific (e.g., number of Gs in the G-runs, the nucleotide composition in the loop). The reviewer is very much correct that longer G-runs (e.g., GGGG) are stronger and additional G-runs can serve as “spare tyres” as well. However, this new preprint also shows that excessively long or overlapping G-runs may support multiple competing rG4 topologies, which in contrast, may destabilize the structure (PMID: 23683360). Thus, while additional G-runs can

act as backup sites for rG4 folding, their contribution to increased stability is likely context specific. We updated the main text to make this point clearer.

- Fig 3B : should be easy to add the G4 candidate sequence- not simply the SNP.

We thank the reviewer for this helpful suggestion. In response, we updated the figure to show both the portion of the 5'UTR sequence where the putative rG4 regions are and the predicted impact of the SNP on rG4 formation.

These new visualizations are designed to complement our structural interpretations and also address Reviewer 2's related request for more intuitive structural representations of the EPN3 and MSH6 variant effects. Specifically, we now include illustrations that highlight the guanine runs and the location of the variant within the rG4 motif.

- Fig 4B: I am surprised by the near 0 / flat line CD spectra of the non-G4 forming sequence (dark blue) at 25°C: single stranded nucleic acids generally give a signal at room temperature due to base stacking – a flat line is generally observed only at high temperature.

This is very true - We sincerely thank the reviewer for their careful observation and regret that this escaped our attention. The reviewer's comment prompted us to check the MSH6 WT CD raw data, which was consistent across replicates. However, when we went back to the original RNA oligo, we noted that by TBE-UREA PAGE that MSH6 WT RNA was virtually undetectable, while the other three RNAs (MSH6 mutant, EPN3 WT, and EPN3 mutant) were all readily detected.

We had both MSH6 WT and MSH6 Mutant resynthesized by IDT and ensured the correct concentration of the newly synthesized RNAs by both TBE-Urea gel and UV-Vis spectrophotometry.

We have repeated the CD experiments using these newly synthesized RNAs and indeed, we now find a signal for this RNA as expected by the reviewer. We have updated the figure in the main results to reflect this change. We apologize for this oversight in the original submission.

Figure 1

Figure 1: Comparison of RNA QC check by TBE-UREA PAGE. As shown on the left, MSH6 WT RNA was fully absent (potentially due to degradation and/or synthesis issues). RNAs were resynthesized and QCed (right), and as shown, were of good quality.

Figure 2

Figure 2: CD spectra were re-collected on newly synthesized WT and mutant MSH6 RNAs. As shown, the WT now displays the expected signal, consistent with the reviewer's comment. The mutant signal was also reproducible using a new batch of RNA, in agreement with the original figure.

Reviewer #2 (Remarks to the Author):

The authors introduce G4mer, an RNA language model based on the transformer architecture, designed to predict RNA G-quadruplexes (rG4s) and assess the effects of genetic variants on rG4 formation. The study integrates computational modeling, experimental validation, and disease association analysis to explore rG4s' functional and structural roles. The authors claim that G4mer outperforms existing tools, such as CNN-based G4detector, scoring-based cGcC, and G4Hunter, particularly on longer transcriptomic sequences and datasets from diverse experimental protocols. The model was further applied to study the functional significance of rG4-altering variants in diseases, including breast cancer, using genome-wide association studies (PheWAS) and experimental approaches like dual luciferase reporter assays and circular dichroism spectroscopy. I have the following major concerns regarding the work.

We sincerely appreciate the reviewer's thoughtful summary of our work and their recognition of the multiple components of our study, including the integration of computational modeling, experimental validation, and disease association analyses. We are grateful for the opportunity to address the reviewer's concerns and to further clarify our methods and findings. Below, we provide detailed responses to each of the points raised.

1. In the initial experiments involving binary classification tasks with rG4-seekers, although G4mer outperforms G4Detector, both achieve high AUC scores exceeding 0.9. This suggests that detecting or distinguishing rG4 from other structures may be relatively straightforward. However, when transitioning to subtype classification, G4mer demonstrates significantly superior performance compared to G4Detector. If the finer-grained classification is more significant, the subsequent analysis pertaining to subtypes seems to receive less attention. It appears that the binary classification version suffices for the ensuing experiments.

We thank the reviewer for pointing out the lack of subsequent subtype-specific analysis. In general, we believe that while binary classification achieves high performance, the improvement in fine-grained subtype classification is both more technically challenging and biologically informative. rG4 subtypes differ in their structural stability, folding energetics, and potential regulatory interactions (Kwok et al., 2016; Chambers et al., 2015). As such, subtype classification could offer insight into the specific roles that rG4s play in processes such as splicing, mRNA localization, or translational control.

Following the reviewer's comment, we added subtype-specific analyses. First, we analyzed the transcriptome-wide distribution of predicted rG4 subtypes, comparing them to the training distribution and to the distribution of cases where mutations are predicted to disrupt rG4s. We found significant distribution shifts across subtypes ($p < 10e-30$, Chi-square test). Specifically, the training data contains more representation from the bulges, canonical, long loop, and two-quartet categories, while transcriptome-wide predictions were dominated by "Potential G-quadruplex & G \geq 40%" subtypes. This shift was also clear for cases where mutations disrupted rG4s. The exact reason compared to the distribution of subtypes in the training data is unknown and could stem from technical experimental biases in rG4-seq or modeling biases.

Nonetheless, the enrichment for noncanonical sequence features in transcriptome and mutation-disrupted rG4s points to the biological relevance of subtype classification and the importance of modeling diverse G4 structural configurations beyond canonical motifs.

Next, to investigate the functional relevance of distinct rG4 subtypes, we performed a subtype-specific analysis of rG4-breaking effects. Specifically, we asked whether certain rG4 subtypes are more susceptible to disruption by naturally occurring single-nucleotide variants (SNVs). To do this, we first identified putative transcriptomic rG4 sequences overlapping with gnomAD variants that were confidently predicted by G4mer (score > 0.7) and were strongly disrupted by the variant (mutant score < 0.3), following the same filtering criteria used in our rG4-altering variant analyses. We then assigned each high-confidence rG4 a subtype using our multiclass G4mer model and quantified the degree of disruption using the change in G4mer score (Δ G4mer) between the wildtype and mutant sequences.

Interestingly, as we see above, rG4-breaking effects varied substantially between subtypes. Two-quartet and bulge subtypes tended to show more severe decreases in G4mer scores upon mutation, suggesting that these configurations may be more sensitive to single-nucleotide variation. Canonical and long loop subtypes, by contrast, exhibited somewhat milder destabilization, consistent with their expected higher structural stability. Notably, the overall trend of mutational sensitivity across subtypes mirrors the known hierarchy of rG4 structural stability reported in previous studies (Chambers et al., 2015), reinforcing that G4mer captures biologically meaningful differences between rG4 subtypes.

To further assess the functional relevance of rG4 subtype stability, we analyzed the mean allele frequencies (MAF) of gnomAD variants that disrupt the GG dinucleotide context of putative rG4 structures, stratified by subtype.

In the figure above, we see that variants that disrupted potentially shorter G-runs, such as two-quartet and G-triplex subtypes, have higher MAF compared to variants disrupting other subtypes, suggesting weaker negative selection.

Taken together, we believe these new analyses provide functional context to the importance of subtype classification: rG4 subtypes differ not only in structural configuration but also in their mutational robustness and potential selective pressures in the human population. We thank the reviewer again for suggesting expanding the subtype analysis. Accordingly, we have updated the manuscript with a new main figure (Figure 3), a new supplementary figure (Figure S3), and expanded the corresponding Results and Discussion sections.

2. Although the EIG analysis provides insights into the importance of flanking sequences, other interpretable features, such as specific sequence motifs or secondary structures, are not deeply explored.

We appreciate the suggestion to further examine the biological and structural context of rG4 flanking sequences. To address the reviewer's comment regarding the exploration of additional interpretable features beyond the EIG analysis, we have now performed a detailed motif

enrichment analysis focusing on the UTR flanking regions of sequences containing canonical rG4 motifs that were predicted to fold versus those that do not.

Specifically, we scanned the 5'UTR and 3'UTR sequences separately using the CISBP-RNA motif database and FIMO (from MEME suite 5.5.7 (link)). For each motif, we calculated the fraction of sequences in which the motif was present in both the rG4-forming (target) group and a non-forming control group. We then computed the fold enrichment as the ratio of the motif occurrence fraction in the forming group relative to that in the nonforming group, and transformed this ratio to a \log_2 scale.

To assess significance, we first computed Fisher's exact test p-values for all motifs with sufficient hits (at least five total hits) across groups. These were adjusted for multiple testing using the Benjamini–Hochberg procedure. Only motifs passing the FDR threshold ($q < 0.05$) were then filtered by effect size (\log_2 fold enrichment > 1) for reporting. The result of this analysis appears now as a new Figure 3f and is pasted below.

As can be seen in the figure below, the motif mapping to hnRNP F/H exhibited the highest \log_2 fold enrichment ($\sim 17 \log_2$ fold enrichment), indicating that it is overrepresented in the target set relative to controls. Notably, hnRNP F has been shown to bind RNA G-quadruplexes and regulate alternative splicing in a G4-dependent manner, independent of linear G-runs (Huang et al., 2017). This suggests that rG4 formation—and not merely G-run presence—can modulate RBP binding and downstream processing such as splicing. This finding is in line with our finding that the flanking regions of canonical rG4 motifs predicted to fold into rG4 structures are significantly enriched in G-rich sequences compared to those predicted not to fold. This observation not only complements the EIG analysis, underscoring the importance of G-richness in the flanks, but also suggests that the formation of rG4s may be intricately linked to RBP regulation. For instance, the pronounced enrichment of the hnRNP F/H motif, which preferentially binds G-run sequences, implies that rG4s (or a distinct subtype of rG4s with longer loops and G-rich flanks) might modulate the binding of those RBPs. Whether the enriched G-rich flanks primarily facilitate the formation of a specialized rG4 subtype and/or participate in recruiting RBPs is yet to be explored.

In addition to the enrichment of the hnRNP F/H motif, we observed significant enrichment of motifs associated with SRSF4/SRSF6, with \log_2 fold enrichments of approximately 16.7. The SRSF family proteins (SRSF4 and SRSF6) are well-established splicing regulators that bind to exonic splicing enhancers and modulate splice site recognition, Their binding is often influenced by the local RNA structure, and previous studies have shown that changes in RNA secondary structure can alter the activity of SR proteins, thereby impacting alternative splicing decisions. Moreover, recent work by NIU et al., Nucleic Acids Research, 2024, provides compelling evidence that SRSF1, an SR protein, can actively unfold RNA G-quadruplex structures. In their study, SRSF1 binding leads to the destabilization of rG4 structures, thereby modulating RNA secondary structure and influencing downstream processes such as splicing and mRNA translation. This observation is particularly relevant to our findings: we observe significant enrichment of SRSF4/SRSF6 motifs in the flanking regions of rG4-forming sequences, as well

as moderate enrichment of SRSF1 motifs in both the 5' and 3' UTR contexts. The mechanism proposed for SRSF1 suggests a broader role for SR proteins in regulating rG4 stability.

We also observed significant enrichment of motifs associated with RBM4/RBM4B with \log_2 fold enrichments of approximately 16.2. Indeed, in the study “G-Quadruplex Regulation of VEGFA mRNA Translation by RBM4” (Niu et al., 2022), the authors demonstrate that RBM4 directly binds to G4 structures within the 5'UTR of VEGFA mRNA, thereby modulating its translation. Notably, the paper shows that altering RBM4 expression levels affects the formation or stability of these G4 structures, ultimately influencing VEGFA expression. This finding underscores a functional role for RBM4 in G4-mediated translational regulation. In our analysis, the marked enrichment of motifs associated with RBM4 (and its paralog RBM4B) in rG4-forming 5'UTR sequences supports the notion that RBM4 may similarly interact with G-rich flanking regions to modulate RNA processing events such as alternative splicing and translation.

The 3'UTR list of significantly enriched motifs also shows SR proteins like SRSF1 and SRSF2, as well as hnRNP F/H, which we also saw in the 5'UTR list. Additionally, we found the highest enrichment for ELAVL (Hu) proteins (e.g., ELAVL2, ELAVL3). ELAVL proteins classically bind AU-rich elements and are implicated in mRNA stabilization. Interestingly, our EIG analysis highlighted high attribution scores of AU dinucleotides in the flanking regions of rG4 motifs. Consistently, our motif enrichment analysis of 3'UTR sequences revealed significant enrichment of ELAVL-associated motifs, and since ELAVL proteins are well-known to bind AU-rich elements, this suggests that AU-rich flanks may recruit ELAVL proteins to modulate mRNA processing.

In summary, our findings reinforce the hypothesis that the flanking regions of canonical rG4 motifs are not only critical for rG4 formation—as supported by our EIG analysis—but may also serve to modulate RBP binding and potentially the unfolding activity of the structure. These insights open exciting avenues for future studies investigating rG4s as modulators of RNA processes and their functional subtypes.

Finally, we included the RBP motifs mentioned above, where we see G-richness for HNRNPF and SRSF4, and U-richness of ELAVL1, all of which appear as significant motifs in both 5' and 3' UTR flanking regions of rG4s. We have also added these alongside the RBP motif enrichment bar plots.

We thank the reviewer for the suggestion to further pursue this direction of investigation. All of the above results are incorporated in the new Fig. 3.

3. On the other hand, EIG is not suitable for analyzing large-scale language models due to significant deviations in the integrated gradients of the input. In my view, the attributions from this method may not be sufficiently appropriate to draw the conclusion that "sequences flanking the G-tracts do indeed contribute to the rG4 predictions." I suggest utilizing a perturbation-based method directly to obtain input attributions, similar to the approach you employed in predicting the rG4-altering effects of single nucleotide variants. You can alter the nucleotides in the flanking region, and observe the changes of the prediction.

We thank the reviewer for this thoughtful comment. We agree that interpreting attributions in large-scale language models presents specific challenges, particularly due to the complex interactions introduced by embedding layers and self-attention mechanisms. While methods like Integrated Gradients (IG) have known limitations in transformer-based architectures, we believe that Enhanced Integrated Gradients (EIG) provides useful insights in our case. Specifically, three key elements distinguish EIG: that the integration can be performed over latent space representations rather than the original space; that the reference point can be set to a more biologically meaningful one and not necessarily a single point, such as clusters of "negative" samples in latent space; and that EIG includes a statistical significance test for enriched features for a class of points of interest. Indeed, EIG has previously been shown to offer biologically meaningful attributions in genomic deep learning models, including convolutional architectures (Jha et al, Genome Biology 2020). In the context of this work, we first assessed EIG-based attributions using the aforementioned statistical testing for the entire rG4 suspected regions. This yielded, as expected, the known G4 tracts. But the same statistical test pointed to short motifs/k-mers in the adjacent flanking regions that also consistently received signal across many examples, suggesting that these regions may contribute to rG4 predictions.

Nonetheless, we appreciate the reviewer's suggestion to complement this analysis with perturbation-based attribution. As recommended, we have now implemented a perturbation analysis in which we systematically altered nucleotides in the flanking regions and measured the resulting change in rG4 prediction. This approach provides a more direct and

model-agnostic evaluation of sequence importance. These results are now included in Supplementary Fig. S4c, which is referenced in the corresponding section of the main text. Details of the analysis are provided below.

We performed a perturbation analysis to further validate and complement our EIG interpretation, specifically examining how nucleotide substitutions in the flanking regions affect G4mer predictions. Using sequences with high-confidence rG4 predictions (G4mer scores > 0.9), we systematically mutated guanines (G) and cytosines (C) within the flanking regions and assessed the impact on G4mer scores. Perturbing guanines to cytosines G → C consistently caused the largest reduction in predicted scores compared to other perturbations, highlighting that guanines in the flanking regions contribute to stabilizing the predicted rG4 structures. Conversely, substitutions of cytosines to guanines C → G led to the largest increase in the predicted rG4 scores compared to the other perturbations, reinforcing the importance of guanine content around the rG4 motif. These perturbation results support our initial interpretation derived from the EIG analysis, emphasizing that the nucleotide composition of flanking regions—particularly the enrichment of guanines—has the greatest contribution to rG4 predictions by G4mer, and that the presence of cytosines in the flanking regions could weaken the rG4.

We thank the reviewer for encouraging us to try an alternative interpretation strategy, which further supported our conclusions about the importance of flanking regions for rG4 formation.

4. It appears that your experiments concerning rG4 alterations primarily focus on single nucleotide variants. Have you considered the impact of multiple nucleotide variants? In the section analyzing PMBB variants, do they involve alterations of multiple nucleotides? Perhaps it would be beneficial to delve deeper into discussions and analyses on natural mutations rather than artificial negative selections. Understanding the relationships between nucleotides is crucial and prevalent in nature.

The reviewer raises some good questions here. To clarify, our analysis focuses on naturally occurring single-nucleotide variants (SNVs) derived from whole-genome sequencing data in the Penn Medicine BioBank (PMBB), not artificially introduced mutations.

We agree that multi-nucleotide variants (MNVs), particularly those that are phased and occur on the same haplotype, may also impact rG4 structure. In this study, we focused on SNVs because they are the most common and best-characterized class of variation in large-scale population datasets such as gnomAD and PMBB, and they are well-suited for phenotype association studies like PheWAS. At present, most phenotype-genotype association methods, including PheWAS, are built to operate on individual variants, and reliable phasing information for adjacent variants is often not available in population-scale biobanks.

That said, we recognize the importance of studying combinations of nearby variants, particularly in structured regions like rG4s, where cooperative effects among nucleotides may influence folding. Notably, G4mer is a variant-agnostic sequence model that takes raw nucleotide sequences as input. As such, it is readily extensible to analyzing MNVs and even larger haplotype blocks, provided phased sequence information is available. This flexibility opens up new opportunities for studying how combinations of variants influence rG4 formation, especially in regions with complex or overlapping G-runs.

To emphasize the point that the variants we tested are naturally occurring mutations, we have clarified in the revised text that the variants analyzed are SNVs from patient-derived data, and have added a comment on the potential value of extending G4mer to phased or multi-nucleotide variant analysis in future work.

5. While the study validates rG4-altering variants in two genes (EPN3 and MSH6) for breast cancer, it does not extend this approach to other disease-associated genes or phenotypes. A broader experimental validation across multiple variants would strengthen the claims.

We fully agree that expanding experimental validation to additional disease-associated variants and phenotypes would further strengthen our findings. However, given the low-throughput nature of functional assays, such as dual luciferase reporter assays and circular dichroism spectroscopy, experimentally validating a broad set of variants at scale is not currently feasible.

In this study, we prioritized two rG4-altering variants in EPN3 and MSH6 as a proof of concept to demonstrate the impact of predicted rG4-disrupting and rG4-forming variants on post-transcriptional regulation.

To ensure the robustness of our findings, we validated these effects in multiple independent cell lines with both technical and biological replicates. While our *in cellulo* dual-luciferase experiments are low-throughput, they offer key advantages by enabling direct functional assessment of long, full-length 5'UTR sequences in the context of a reporter gene (i.e., analogous to their native regulatory function). This enables more precise quantification of rG4-mediated regulatory effects. Although validating additional variants would further strengthen our findings, our approach ensures biologically relevant functional evidence paired with structural evidence, which remains critical for interpreting rG4-associated regulatory mechanisms. Future studies integrating high-throughput functional assays with *in cellulo* validation, which is currently still limited, could provide a broader yet mechanistically grounded understanding of rG4-mediated regulation across multiple variants and phenotypes.

Additionally, we view this work as an initial demonstration of the predictive power of G4mer, with the intent to extend its application to other disease-relevant rG4 variants in follow-up studies, beyond the scope of the current manuscript. Finally, it is worth pointing out in this context to a recent preprint from the Dominguez lab (Martyr et al., 2025), which includes thousands of variants around a selected set of several rG4 sequences. The results in this study clearly support our computationally derived conclusion regarding strong context-specific effects of variants in regions flanking the core G-runs, the loop length/composition, etc. We have added information in the Discussion section and appreciate the reviewer's suggestion in guiding future directions.

6. The model is trained exclusively on human transcriptomic data. It is unclear whether G4mer could generalize to predict rG4s in other organisms, such as plants or bacteria, where rG4s also play regulatory roles.

The reviewer raises an important question here. It is true that G4mer is currently trained exclusively on human transcriptomic data, as our primary focus was to investigate rG4 formation and variant effects in the context of human gene regulation and disease.

We agree that evaluating G4mer's ability to generalize to other organisms — such as plants or bacteria, where rG4s also play important regulatory roles — is an exciting direction for future work. However, we also recognize that cross-species generalization may be non-trivial due to species-specific differences in sequence composition, structural stability, and transcriptomic context. Specifically, some prior studies have shown that RNA regulatory elements and structural motifs can diverge significantly across species, often requiring model retraining or domain adaptation to achieve reliable performance (Harris et al., 2024; Karollus et al., 2024; Ferhadian et al.). Nonetheless, to facilitate such investigations, several datasets are available, providing transcriptome-wide rG4 mappings in non-human species. For example, the G4Atlas database includes rG4-seq and chemical probing data for *Arabidopsis thaliana* and *Escherichia*

coli. Leveraging these datasets, future studies could explore whether models like G4mer can be adapted to predict rG4s in diverse species through transfer learning or fine-tuning approaches.

As our focus is on understanding human rG4s, genetic variants, and their associations with human disease, developing a model that generalizes across species is beyond the scope of the present study. However, we have now added a discussion of this limitation and potential future extensions in the revised manuscript.

7. While G4mer achieves high-performance metrics, the reliance on a single experimental dataset (rG4-seq) raises concerns about overfitting to specific experimental artifacts. The authors should have included independent validation datasets from additional sources, such as SHAPE-MaP or other RNA structure-seq methods.

We appreciate the reviewer's concern regarding potential overfitting to a single experimental dataset (rG4-seq). We agree with the reviewer that since G4mer was trained on rG4-seq, validating its generalizability is essential. To alleviate this concern, we previously included a comparison of G4mer's performance (along with other models) on an independent dataset from G4RNA, as presented in the Results section. G4RNA is a comprehensive database of experimentally validated rG4 sequences collected from 24 different experimental protocols across multiple studies, ensuring a diverse validation set beyond our training data. The validation results on G4RNA, shown in Figure 1d, demonstrate that G4mer outperforms existing models, particularly for longer sequences, many of which originate from native transcripts. This evaluation was conducted to ensure that G4mer is well-suited for predicting rG4s in long transcriptomic sequences.

Additionally, in Figure 1e, we show that G4mer achieves higher performance across sequences derived from the five most common experimental techniques in G4RNA, reinforcing its robustness across multiple methodologies. These results indicate that while G4mer was trained on rG4-seq, it did not overfit a single experimental protocol and generalized well to independent validation datasets.

The reviewer also asked specifically about structure probing data. We want to clarify that the G4RNA database includes sequences derived from SHAPE-based probing methods (SHAPE, SHALiPE) and DMS-based approaches (DMS, DMSLiPE), which are widely used in RNA secondary structure characterization. Additionally, G4RNA contains data from CMCT probing and RNase-based structure probing methods (RNase T1, V1, and T2), among others, further diversifying the experimental sources validating G4mer’s predictions. To illustrate this diversity, in the Results section, we reference Supplementary Figure S2a (shown below), which summarizes the range of sequences in G4RNA, spanning various transcriptomic contexts, sequence lengths, and both natural and synthetic sequences.

To improve clarity, we have updated the Results section to further emphasize the diverse experimental techniques represented in G4RNA and their role in validating G4mer’s generalizability. Additionally, we have generated a new supplementary figure (Fig. S1c), included below, which expands the performance comparison of G4mer and other models from the top 5 (as shown in Fig. 1e) to the top 15 experimental protocols in the G4RNA database. This expanded analysis includes rG4 sequences validated through SHAPE, SHALiPE, DMS (DMS, DMSLiPE), and other structure-probing techniques, demonstrating G4mer’s strong performance across varied experimental contexts. We selected the top 15 protocols to capture

the most well-represented experimental datasets ($n \geq 5$ sequences per protocol), but we made an exception for certain SHAPE-based protocols that had fewer sequences, in response to the reviewer's request to assess SHAPE-based validation.

We thank the reviewer for this valuable feedback.

8.Lack of clinical validation metrics: The study applies G4mer to predict rG4-breaking variants in disease-associated genes, but the clinical implications of these predictions are not quantified. For example, how well does G4mer distinguish between pathogenic and benign variants in ClinVar? Are there false positives in the predicted rG4-breaking variants from gnomAD?

The reviewer raises several important points. Regarding clinical applications and ClinVar: We appreciate the reviewer's suggestion to assess our predictions with respect to clinically annotated variants. In response, we downloaded the ClinVar dataset (GRCh38 build) from the NCBI FTP site (ftp://ftp.ncbi.nlm.nih.gov/pub/clinvar/vcf_GRCh38/) and extracted variants located in the UTRs. We then used our G4mer model to generate both wildtype and mutated rG4 predictions (i.e., G4mer scores) for these variants. Specifically, we focused on variants with clear clinical significance annotations (Benign vs. Pathogenic) and a high wildtype rG4 score (>0.7) to capture putative rG4 regions. For each unique variant-transcript mapping, we retained the most disruptive effect (the row with the minimum $\Delta G4mer$, defined as mutated score - wildtype score).

As shown in the figure (below), we observed that rG4-breaking variants—defined by negative Δ G4mer scores (mutated G4mer score – wildtype score)—were more pronounced among ClinVar variants annotated as pathogenic. To assess this systematically, we performed a one-sided Mann-Whitney U test comparing the distributions of Δ G4mer scores between pathogenic and benign variant groups. In the 5'UTR, pathogenic variants showed significantly more negative Δ G4mers compared to benign variants (***, $p = 5.21 \times 10^{-9}$), supporting potential link between rG4 destabilization and clinical pathogenicity.

We extended this analysis to the 3'UTR and found a consistent trend: pathogenic 3'UTR variants have significantly stronger breaking effects (greater decrease in G4mer scores) compared to benign variants (***, $p = 1.03 \times 10^{-14}$, one-sided Mann-Whitney U test), indicating a similar pattern of rG4 disruption in the 3'UTR.

The above results point to the potential utility of including G4mer in assessing pathogenic variants, for example, in ensemble-based pathogenicity scoring methods. For example, at a $\Delta G4mer$ threshold of -0.2 (mutated score - wildtype score), we observe an increase in the odds of a variant being pathogenic among rG4-breaking variants relative to other variants within predicted rG4 regions (wildtype G4mer score > 0.7). In the 5'UTR, the odds increase from 0.15 (403 / 2628) to 0.27 (98 / 369), yielding a 1.74-fold enrichment. In the 3'UTR, the odds increase from 0.32 (1705 / 5328) to 0.42 (565 / 1336), corresponding to a 1.32-fold enrichment. At a more stringent threshold of -0.5, the odds increase to 0.80 (31 / 39) in the 5'UTR and 0.48 (309 / 646) in the 3'UTR, corresponding to 5.20-fold enrichment (p-value $2.30e-10$, binomial test) and 1.49-fold enrichment (p-value $8.92e-9$, binomial test), respectively. These results suggest that G4mer-predicted rG4-breaking events are enriched in pathogenic variants and may offer complementary information for ensemble pathogenicity scoring. However, we caution that ClinVar itself is a biased and incomplete dataset, and further studies are needed to evaluate the clinical utility of rG4-based features. Regardless, we believe applying G4mer can help suggest a mechanistic interpretation to such variants.

A second important point raised by the reviewer regards the potential for false positives among the predicted rG4-breaking variants from gnomAD. Accurately quantifying false positives in a population database such as gnomAD is challenging because a comprehensive “ground truth” for rG4 structure formation in every UTR is not available. To mitigate this we performed the following. First, our analysis applies stringent filtering criteria: we restrict predictions to variants that occur in putative rG4-forming regions (wildtype G4mer score > 0.7) and applied strict quality control (QC) procedures—similar to the ones used in the gnomAD variant filtering pipeline—to ensure that we only include high-quality, reliable variants in our study. Hence, we filtered out low-confidence or potentially erroneous variant calls using criteria comparable to those used by gnomAD. Second, we estimated the false positive rate for our G4mer predictions by benchmarking against rG4-seq data in highly expressed genes. In this approach, regions of the genes that did not yield any rG4 detection in rG4-seq are considered “true negatives.” Using this

true negative set, we calculated the false discovery rate (FDR) and obtained a value of 0.01863. This shows that while G4mer might have some false positive predictions, it has a low FDR rate overall of less than 2% thus should serve as a reliable tool for transcriptome-wide rG4 prediction.

To conclude, given that there are no HT measurements for rG4 structures with gnomAD variants, the above approach of using rG4-seq is currently our best available experimental method for detecting rG4 structures in order to estimate the false positive rate. We acknowledge it has limitations. For example, rG4-seq may not capture all rG4s (or have subtypes biases) due to sensitivity issues, transient formation, or variability between cell populations. As a result, some rG4s might be missed (false negatives), which could lead to an overestimation of the false positive rate when comparing our computational predictions to the rG4-seq data. We discuss these limitations in the revised manuscript.

We thank the reviewer for suggesting analyzing G4mer's FDR to further strengthen the credibility of the model quantitatively.

9. While the authors emphasize the importance of longer rG4s, the biological mechanisms underlying their functional significance remain speculative. Experimental validation to confirm whether longer rG4s are more prone to disruption by variants would add value.

We appreciate the reviewer's thoughtful suggestion regarding the biological relevance of rG4 length and its relationship to variant sensitivity. While we agree that this is an important and interesting direction, a detailed mechanistic investigation of rG4 length and its susceptibility to disruption by genetic variants is beyond the scope of the current study, which is primarily focused on developing and validating a predictive model for rG4s and variant effects transcriptome-wide. That said, we point to a new high-throughput experimental study investigating exactly this question in a pre-print from the Dominguez lab (Martyr et al, 2025). This study assessed the length and composition of the spacer regions between the G-runs, and the results in this study are in line with our computational predictions. We view further investigation of the role of sequence length and composition in the formation of G4s as an exciting avenue for future work and are actively exploring it with the Dominguez lab.

10. In terms of the structural validation of the two specific rG4 variants from EPN3 and MSH6, providing the specific secondary structure would enhance clarity and intuition. Additionally, the manuscript lacks any structural visualization of these variants, which diminishes the overall intuitiveness of the work.

We thank the reviewer for this suggestion. We agree that including structural visualizations can enhance the clarity and interpretability of our findings. In response, we explored several established RNA structural prediction tools to generate visualizations for the rG4-forming and rG4-breaking variants in EPN3 and MSH6.

We first considered RNAfold, a widely used tool for predicting RNA secondary structures based on minimum free energy (MFE) conformations. While RNAfold includes an option to incorporate G-quadruplex formation into its energy model, this implementation treats G4s as an energetic bonus rather than explicitly modeling their structural topology. As a result, the predicted secondary structure output remains canonical—composed of standard base pairs—and does not provide a graphical representation of G4 formation (e.g., G-quartet stacking or G-run coordination). This limits its utility for visualizing the structural impact of variants on rG4 folding. Therefore, despite enabling the G-quadruplex option, RNAfold is not suitable for generating accurate or intuitive visualizations of rG4 structures.

We also evaluated Rosetta FARFAR2, a state-of-the-art tool for de novo RNA tertiary structure prediction. While FARFAR2 has demonstrated success in modeling a variety of complex RNA folds, including hairpins and multi-way junctions, it is not equipped to predict G-quadruplex formation from sequence. This is due to limitations in fragment assembly and scoring functions, which do not account for the unique geometry, base-stacking, and cation-stabilized interactions of rG4 structures. To our knowledge, no prior studies have used FARFAR2 to model RNA G-quadruplexes.

We also considered AlphaFold3, which has recently incorporated RNA-protein complex modeling capabilities. However, the model's RNA folding component is still in early stages, and support for RNA-only secondary and tertiary structure prediction—especially for noncanonical motifs like G-quadruplexes—is not yet available or publicly usable for the purposes of standalone rG4 structure inference.

Given these limitations, and in the absence of an available RNA structure prediction method that can accurately and visually model rG4 folding from sequence—including stacking geometry and G-quartet interactions—we chose not to include structural visualizations in the revised manuscript. We believe that including potentially misleading or oversimplified depictions would not enhance interpretability and could risk misrepresenting the structural impact of the variants. We look forward to incorporating such visualizations in future work as structure prediction tools for rG4s mature.

11. I got "Repository not found" when opening the code link.

We apologize for any inconvenience caused by the reported error message. It is possible that there was a temporary permission issue. We have confirmed that the repository is publicly accessible, and other reviewers seem to have accessed it without any issue. We kindly invite the reviewer to try accessing the repository again using the following link: <https://bitbucket.org/biociphers/g4mer/src/main/>. If issues persist, please let us know via the editor, and we would be happy to assist further. Again, we apologize for the inconvenience it might have caused.

12. Typos:

page9, "<=28nt" seems to be ">=28nt"

We thank the reviewer for catching this typo. We have corrected it in the text to accurately reflect that long rG4 sequences are defined as ≥ 28 nt. The updated sentence now reads:

"Next, we investigated the MAF of variants located within short (≤ 22 nt) and long (≥ 28 nt) rG4 sequences across both 5'UTR and 3'UTR regions."

Reviewer #3 (Remarks to the Author):

Review Nature Communications

Thank you for allowing me to review this paper. It is very strong but with a few caveats for improvement. Please see below for my comments.

We sincerely appreciate the reviewer's time and effort in evaluating our work. We are grateful for the thoughtful feedback and constructive suggestions, which have helped us improve the manuscript. Below, we address each comment in detail.

"We first evaluated its performance against current state-of-the-art methods across diverse data and experimental techniques, showing that it significantly outperforms existing methods"

Comment: Diverse data like what? Do you mean the makeup of the individuals in the dataset or just a collection of different datasets? This might be misleading language.

Thank you for this helpful clarification request. In this context, we use "diverse data" to refer to the range of dataset sources, sequence lengths, transcriptomic contexts, and experimental techniques used in our evaluation. We recognize that the original phrasing may have been ambiguous, and we have revised the manuscript to clarify the specific aspects contributing to this diversity. Additional discussion of this point, including a breakdown of dataset sources and sequence types, is provided in our response to Reviewer #2, point 7. We appreciate the reviewer's suggestion to improve the clarity of this phrasing.

"On this task, G4mer outperformed G4Detector, obtaining an accuracy of 0.95 and a receiver operating characteristic area under the curve (ROC-AUC) score of 0.98, compared to an accuracy of 0.92 and an AUC of 0.97 for G4Detector (Fig.1c)." And related metrics throughout.

Comment: Accuracy is a notoriously problematic measurement for model validation in classification- You include ROC here so you've covered sensitivity (aka recall) and specificity, but adding precision-recall AUC I think makes a stronger argument instead of just ROC and accuracy. You've already calculated this in the supplement so it should be an easy add.

The reviewer raises an important point regarding measures of performance. We agree that accuracy alone can be a limited metric for model validation, particularly in classification tasks with imbalanced datasets. While we included ROC-AUC, which captures both sensitivity (recall) and specificity, we acknowledge that Precision-Recall AUC (PR-AUC) provides additional insight, especially in cases where class distributions are skewed. To address this, we have updated the Results section to include PR-AUC values alongside the ROC-AUC and accuracy metrics that we previously had to provide a more comprehensive evaluation of G4mer's performance.

Below, we provide an updated plot incorporating PR-AUC for both rG4 prediction and rG4 subtype prediction. We have updated the existing Figure 1c in the main manuscript accordingly.

“In this data, the authors categorized each sequence into one of eight labels, ranging from canonical to known noncanonical subtypes such as long loops, bulges, and two-quartets.”

Comment: Is this a standard categorization technique or was this using content area expertise to develop the classification? Be more specific about how this classification was done.

The reviewer's point is well made. Indeed, the rG4 classification procedure was not clarified in the original submission. The classification of rG4 sequences into eight subtypes follows prior literature, particularly Kwok et al. (2016) and Chow et al. (2020), which categorized rG4s using structural motifs and stability features. The rG4-seeker dataset, which we used to train and test G4mer, applies these classification rules using a hierarchical approach, prioritizing sequence motifs with higher predicted stability through existing literature studying G4 sequence stability.

To improve clarity, we have:

1. Updated the Methods section (rG4 formation and subtype classification datasets) to provide additional details on how rG4-seeker applies these classification rules.

2. Revised the Results section to explicitly reference prior work supporting this categorization approach and the methods section for more detail.

We appreciate the reviewer's suggestion to enhance clarification regarding the classification methodology.

Comment: There is good detail here regarding the training and evaluation of the model

Thank you for your positive feedback. We appreciate your recognition of the detail provided in the training and evaluation of our model. Ensuring transparency and reproducibility was a priority, and we are glad to hear that this aspect of the manuscript was clear.

"We further filtered samples and variants in PMBB to be included in our analyses. We restricted the analyses to only keep samples which are 2nd-degree unrelated in each ancestry group, resulting in two groups of samples of European (N=29,362) and African (N=10,217) genetic ancestries. Furthermore, exclusion criteria applied to variants include singletons and high missing call rates (exceeding 0.1)."

Comment: It would be beneficial for the authors to describe how they might be able to use this methodology in other populations besides European and African Descent.

We thank the reviewer for this suggestion. To clarify, the methodology we used for filtering samples and variants is not restricted to European and African ancestry groups and can be applied to other populations as long as sufficient data are available. The key preprocessing steps—including identifying 2nd-degree unrelated individuals within each ancestry group and applying variant-level filters for missingness and singleton exclusion—are generalizable to any population with appropriate genotype and ancestry data. The primary reason we focused on European and African descent in this study is due to the composition of the Penn Medicine BioBank (PMBB), where these groups represent the largest available sample sizes (N=29,362 for European and N=10,217 for African ancestry). Other genetic ancestry, such as Asian, is underrepresented in PMBB with N=979, making it challenging to conduct well-powered analyses. However, as larger and more diverse biobank datasets become available, our approach can be directly extended to additional populations.

To clarify this, we have now updated the methods and discussion sections to explicitly explain the choice of ancestries included in the analyses and state that our methodology is agnostic to ancestry and can be applied to other populations when sufficient data is available.

"ICD-10 codes from PMBB samples were mapped to distinct disease entities (i.e. phecodes) via Phecode Map 1.2 using the PheWAS package in R71."

Comment: Spelling error

We thank the reviewer for catching this typo. We have corrected "form" to "from" in the sentence. We really appreciate the careful attention to detail.

“Each phecode was tested for association with each rG4-altering variant using a logistic regression model adjusted for 24 sex, age at enrollment, age-squared, and the first 10 principal components of genetic ancestry. For each variant, ancestry-specific PheWAS was first performed for two major groups: African (AFR) and European (EUR) ancestry.”

I don't understand why you've stratified by genetic ancestry and then recombine using a meta analysis when you've already controlled for genetic ancestry in the regression via principal components. Your beta estimates should already be adjusted in the first stage without stratifying. Provide more justification for doing the strata and why you meta analyzed or else just go with the non stratified phewas on the whole group. If you are going to stratify and recombine, it would be helpful to have the individual phewas results available in the supplement as well.

Thank you for this insightful comment. The decision to stratify by genetic ancestry before recombining via meta-analysis was made to account for potential ancestry-specific effects that may not be fully captured by adjusting for principal components (PCs) alone. While PCs effectively control for population structure at a global level, they may not fully correct for ancestry-driven differences in effect sizes, particularly for variants with different allele frequencies or ancestry-specific linkage disequilibrium (LD) patterns.

By stratifying PheWAS by ancestry and then combining results via meta-analysis, we aimed to:

- Account for ancestry-specific effect size differences, which PCs alone may not fully adjust for. Meta-analysis allows for these differences rather than assuming a single shared effect.
- Prevent ancestry-driven inflation or bias, ensuring that associations are appropriately weighted by sample size and variance rather than being dominated by one population.
- Improve robustness by detecting both shared and ancestry-specific associations, which would be harder to interpret in a pooled analysis.

Our decision for this approach followed established approaches in both genome-wide association studies (GWAS) and phenome-wide association studies (PheWAS). Several studies have demonstrated the advantages of ancestry-stratified analyses followed by meta-analysis. For example, in a multi-ancestry GWAS of major depression (Meng et al., 2024), researchers stratified by ancestry to account for differences in allele frequencies and LD structure before meta-analyzing results. Similarly, the study *Genome-wide association studies in ancestrally diverse populations* (Peterson et al., 2020) discusses the benefits of separately analyzing ancestries to produce robust results that account for genetic differences between populations while still enabling combined meta-analysis. For PheWAS, *A multi-population phenome-wide association study of genetically-predicted height in the Million Veteran Program* (Raghavan et al., 2022) conducted ancestry-stratified analyses in non-Hispanic Black and non-Hispanic White participants before meta-analysis. This approach not only enabled the detection of

ancestry-specific associations but also ensured that signals observed in the meta-analysis were not disproportionately driven by one ancestry group.

As part of our unified analysis pipeline, ancestry-stratified association testing and meta-analysis were performed across all variants. However, we note that some variants are only observed in a single ancestry group. For example, the variant in EPN3, which we report in the paper, is exclusively present in individuals of European ancestry, and thus, results are only available for that group. Per the reviewer's request, we have updated the Methods section to clarify the analysis.

We acknowledge that performing PheWAS on the full cohort without stratification is still a valid approach, and we appreciate the reviewer's perspective on this. We have clarified in the Methods section that different strategies for handling genetic ancestry exist and that the choice depends on the study's goals and the degree of ancestry-related heterogeneity expected.

Overall this is a very good paper but needs clarity on the methods used in phewas. Additionally, many genetics methodologies are validated on non-diverse dataset and have the potential to perpetuate biases. The authors should spend time discussing how one might validate this method further in genetic ancestries besides European and African.

In our study, we focused on European and African ancestry groups due to the aforementioned limitation with the composition of the Penn Medicine BioBank (PMBB), where these groups comprise the majority of available patient samples in the local region. Unfortunately, the number of individuals from other genetic ancestries in PMBB is currently limited, which presents challenges for conducting well-powered analysis (i.e. N=979, 2.2% Asian).

Nevertheless, we completely agree with the reviewer's point, recognizing the importance of ensuring genetic diversity in association studies. We would like to clarify that our framework, which integrates G4mer predictions with PheWAS, does not learn population-specific information. G4mer is a sequence-based model trained without any ancestry labels, meaning its predictions are independent of population structure. Similarly, PheWAS is an association analysis that examines statistical correlations between genetic variants and phenotypic traits within a given dataset, without an inherent training process that could introduce population-specific biases. Therefore, fortunately, our approach does not perpetuate biases arising from non-diverse training data, but is indeed limited in disease association studies by the availability of genetic diversity within PMBB. Hence, our framework of identifying rG4 variants with associations to human diseases using our model G4mer and PheWAS is agnostic to the population data, and can be easily applied to other ancestry groups as more diverse datasets become available, which we are excited to do in the future.

We hope that the updates made in the Methods and Discussion sections provide better explanations in the text and address the valuable insights raised by the reviewer.

We appreciate this valuable suggestion and hope this clarification addresses the concerns.

Reviewer #3 (Remarks on code availability):

I am very pleased with how the authors organized everything in their GitHub repo. The file structure is very intuitive and they provide complete Jupyter notebooks to reproduce their results.

We sincerely appreciate the reviewer's positive feedback on our repository organization and reproducibility. Thank you for your encouraging remarks!

Jul 17th 2025:

Reviewer #1 (Remarks to the Author):

The authors have addressed all my comments in a satisfactory manner. I recommend acceptance.

We sincerely thank the reviewer for their positive feedback and for their thoughtful and constructive comments throughout the review rounds, which helped improve the manuscript. We are happy our revisions have satisfactorily addressed the reviewer's comments.

Reviewer #2 (Remarks to the Author):

Thank you for your response. However, my previous concerns have not been completely resolved.

We thank the reviewer for the careful review and their continued engagement with our work. We provide detailed responses to each outstanding concern below.

For point 1:

There are still some questions regarding the rebuttal's analyses. The authors state that 'Two-quartet and bulge subtypes tended to show more severe decreases in G4mer scores upon mutation'. I can identify the Two-quartet subtypes from the presented figure, but the Bulges in the 5' UTR panel still exhibit very high values.

Thank you for this observation. We agree that the median Δ G4mer score for the Bulges subtype in the 5' UTR is not as negative as that of the Two-quartet subtype and appears relatively higher than its counterpart in the 3' UTR.

However, our original statement was based on the statistically significant difference between each subtype and the Canonical group. Specifically, both the Bulges and Two-quartet subtypes showed significantly greater decreases in G4mer scores compared to the Canonical subtype (adjusted $p \ll 4.4 \times 10^{-4}$, Games–Howell test), as shown in Fig. 3b (also added below for convenience) as asterisks, and described in the Results section. We hope that this clarified the reviewer's question.

Additionally, why is the G-quadruplex not sensitive to the SNV? Considering its name definition requiring four Gs in its structure, one would expect it to be more sensitive?

We are happy to answer this question. The canonical definition of a G-quadruplex involves four G-runs, each made up of at least three guanines – typically defined by the motif $G_{3+}N_{1-7}G_{3+}N_{1-7}G_{3+}N_{1-7}G_{3+}$. However, it is well known that sequences that have this canonical motif sometimes do not form a G4 structure and sequences that do not adhere to this motif definition actually do form a structure (hence the name “non-canonical”).

It follows that the actual sensitivity of a predicted rG4 structure to a single nucleotide variant (SNV) depends on multiple factors beyond this minimal requirement. For example, the G-run length plays an important role. If a G-run contains more than the minimum three guanines, a mutation affecting one of the additional guanines not involved in Hoogsteen bonding may still preserve the structure. Second, while Canonical G-quadruplexes require uninterrupted guanine tracts, other subtypes such as Bulges tolerate interruptions (e.g., GGAG), meaning that a guanine-disrupting mutation might convert one subtype to another, rather than abolishing rG4 formation altogether.

In summary, as expected and shown in our results, many predicted Canonical G-quadruplexes appear more structurally stable, such that an SNV affecting a single guanine may not fully disrupt folding. This buffering effect may lead to smaller ΔG_{4mer} changes upon mutation. In contrast, subtypes like Two-quartet or Bulges may be more structurally constrained or dependent on fewer G runs, making them more sensitive to mutation.

Moreover, I am curious whether G4mer is reliable in reflecting the changes brought about by the SNV.

We thank the reviewer for highlighting the importance of evaluating SNV-induced effects on rG4 formation. In addition to our transcriptome-wide benchmarking and low false discovery rate (<2%), we conducted a focused analysis to assess G4mer's performance on single-nucleotide variants (SNVs) curated from the G4RNA database.

Specifically, we identified 26 sequence pairs that differ by a single nucleotide (1-nt edit distance) and exhibit a binary rG4 label change (i.e., from G4 to non-G4 or vice versa). For each pair, we compared model predictions between the wildtype and mutant sequences. G4mer consistently predicted the correct directional change, assigning lower scores to validated rG4-disrupting variants and preserving correct ranking across these challenging cases. Despite the subtlety of these edits, G4mer achieved 100% directional correctness, outperforming existing tools:

Directional correctness (% of SNV pairs with correct score direction):

G4mer: 100.0%

rG4detector: 73.1%

cGcC: 65.4%

G4Hunter: 65.4%

We acknowledge that because G4mer is trained as a binary classifier, it may not fully capture the quantitative effects of variants on rG4 folding. As a future direction, we plan to fine-tune G4mer in a regression setting using curated perturbation datasets to better predict variant impact on rG4 stability.

Together, these results highlight G4mer's strong sensitivity to SNV-driven disruptions, particularly in preserving directional effects and improving predictive resolution, supporting its utility for fine-grained variant interpretation.

For point 3:

Thank you for incorporating alternative interpretation methods to cross-validate with your EIG; I believe these additions will fortify your findings. One suggestion is to delve deeper into the discrepancies between these two methods.

We thank the reviewer for this helpful suggestion. As the reviewer noted, the addition of complementary interpretation methods helped fortify our findings. In our case, the EIG and perturbation analyses yielded consistent conclusions, both indicating that flanking guanines have positive attribution scores and stabilize rG4 predictions, while cytosines show negative attribution scores and reduce the model's confidence in rG4 formation.

While both methods identify important flanking features influencing rG4 predictions, they offer complementary insights. EIG attributes feature importance by integrating gradients along a path from a reference input (an rG4 motif predicted not to fold) to a target input (an rG4 motif predicted to fold), quantifying how each feature contributes to the prediction. In contrast, perturbation analysis directly measures the model's sensitivity to individual nucleotide substitutions by observing the change in prediction scores. Together, these approaches provide two complementary perspectives: EIG reflects contribution, while perturbation captures sensitivity.

While the Methods section already included a detailed description of both EIG and perturbation analyses, we have now added clarifying sentences in the main text to highlight the distinction between what each method measures. Specifically, that EIG quantifies feature contributions to rG4 predictions, whereas perturbation analysis measures the model's sensitivity to specific nucleotide substitutions in the flanking regions. This addition strengthens the interpretation and reinforces how both methods support the role of specific flanking compositions in rG4 formation. A more in-depth analysis of where EIG and mutagenesis agree/disagree is far beyond the scope of this specific work, and we suspect may be highly context specific given the above.

For point 10:

I concur with your insights on the current structure prediction tools. However, have you considered exploring secondary structure prediction tools? It might offer additional perspectives for prediction comparison. Inviting a visitation of the structure could enhance the paper's presentation.

We appreciate the reviewer's follow-up suggestion to explore RNA secondary structure prediction tools. In our previous revision, we evaluated a range of established secondary and tertiary structure predictors, including RNAfold, Rosetta FARFAR2, and AlphaFold3. While these tools are powerful in many RNA contexts, none currently provide accurate or intuitive modeling of RNA G-quadruplex (rG4) structures, due to limitations especially for non-canonical rG4s.

For example, although RNAfold offers a --gquad option that incorporates G-quadruplex formation into its free energy model (RNAfold manual page), it does so by assigning an energetic bonus to canonical G4 motifs based on G-run composition and short loops (Lorenz et al., 2012), where they confirmed that they restrict themselves to the simpler case of RNA quadruplexes. Hence, this approach does not explicitly model or visualize the rich diversity of rG4 subtypes, such as bulged, long-looped, or two-quartet structures, which are critical for accurate interpretation in our case.

Consequently, while we fully agree with the reviewer that additional visualization from other prediction tools could enhance the paper's presentation, unfortunately we have found that secondary structure visualizations from tools like RNAfold do not meaningfully reflect the structural changes induced by our predicted as well as experimentally validated rG4-disrupting or rG4-forming variants. We believe that including such oversimplified representations would risk misleading interpretations rather than enhancing clarity.

Here, we show an example of RNAfold output taking into account G-quadruplex, for our experimentally non-canonical wild-type and mutated rG4 sequences from the EPN3 gene.

Experimentally validated wild-type rG4 sequence based on Circular Dichroism

Experimentally validated mutated rG4 (GUG to GUU) that disrupts the structure based on Circular Dichroism

As noted earlier, tertiary tools like FARFAR2 and AlphaFold3 lack support for rG4s altogether. At this time, no publicly available tools offer reliable structural visualization of rG4s, especially non-canonical ones, from sequence.

For these reasons, we opted not to include speculative or potentially misleading structural depictions in the manuscript. We fully agree that accurate structure visualization would enhance the presentation, and we plan to incorporate such visualizations in future work as dedicated rG4-aware modeling tools become available.

Minor:

Why is there no median line in the box plot of the purple box in Figure 3b?

We thank the reviewer for pointing this out. The median line is present in the purple box of Figure 3b, but it visually overlaps with the boundary of the interquartile range, which is why it is not distinguishable in the plot. We confirmed this by examining the underlying values: for this group, the 25th percentile, median, and 75th percentile are -0.403, -0.403, -0.1378. We have left the figure unchanged, as this behavior is expected for box plots when the median coincides with a quartile boundary.

Reviewer #2 (Remarks on code availability):

The readme is too simple to set up the environment for me. It's hard for me to run the entire code.

We thank the reviewer for this helpful feedback. We have updated the README to include more detailed setup instructions, including steps for creating the environment, installing dependencies, and running the notebooks to reproduce the figures. We have also added example commands and clarified the expected directory structure and input formats. We hope this improves the usability of the repository.

Reviewer #3 (Remarks to the Author):

The authors did a wonderful job addressing my comments and satisfactorily addressed each of my concerns. No further comments from me.

Code link in bitbucket needs to be addressed though.

We are grateful for the reviewer's positive feedback and are pleased that our revisions have fully addressed their earlier comments. We greatly appreciate their thoughtful input, which helped improve the clarity and rigor of the manuscript. We address the Bitbucket link issue mentioned below.

Reviewer #3 (Remarks on code availability):

The direct link above says "something went wrong" but if you dig into the code base more, you can see the commits etc. The authors need to update this.

We thank the reviewer for pointing this out. We have re-checked the repository link (<https://bitbucket.org/biociphers/g4mer/src/main/>) and confirmed that it is publicly accessible and fully functional. It is possible that a transient Bitbucket error may have caused the “something went wrong” message, as we found that there was a Bitbucket outage for a couple of days in the beginning of this month (<https://bitbucket.status.atlassian.com/history>) . To ensure smooth access, we have double-checked the landing page, verified that the README and code are visible without login, and confirmed that the commit history is viewable. We invite the reviewer to try accessing the repository again and would be happy to further investigate if any issues persist. We would welcome any details on how the error was encountered (e.g., browser used, specific URL followed, etc), which would help us reproduce and resolve the issue. We thank the reviewer again for pointing out this important issue encountered.

August 25th 2025

Reviewer #2 (Remarks to the Author):

Thank you for the revision! It looks nice to me now. I would suggest to make the code into GitHub, which is easier to access.

Thank you for the reviewer's positive feedback on the revision. We're glad to hear the changes have improved the manuscript.

Reviewer #2 (Remarks on code availability):

It's better to make it into GitHub, which is easier to access.

Thank you for the reviewer's suggestion regarding code availability. We appreciate the convenience that GitHub offers for many researchers and are aware that GitHub and Bitbucket are both widely used code repository platforms with similar core features. In our case, we have made the code publicly available via Bitbucket, which importantly supports open access and version control. While specific preferences may boil down to specific features and integration with other tools (e.g., co-pilot for GitHub as it is owned by Microsoft, Jira for Bitbucket as it is owned by Atlassian), this does not affect the accessibility or reproducibility of our work. Importantly, we have ensured that the Bitbucket repository link and clear usage instructions are provided for users and other researchers.